# Dependency of human and murine LKB1-inactivated lung cancer on aberrant CRTC-CREB activation

Xin Zhou[1,2], Jennifer W Li[3], Zirong Chen[1,2], Wei Ni[1,2,4], Xuehui Li[1,2], Rongqiang Yang[1,2], Huangxuan Shen[1,5], Jian Liu[6,7], Francesco J DeMayo[7], Jianrong Lu[2,3,4], Frederic J Kaye[2,8], Lizi Wu[1,2,4]*

[1]Department of Molecular Genetics and Microbiology, University of Florida College of Medicine, Gainesville, United States; [2]UF Health Cancer Center, Gainesville, United States; [3]Department of Biochemistry and Molecular Biology, University of Florida College of Medicine, Gainesville, United States; [4]UF Genetics Institute, Gainesville, United States; [5]State Key Laboratory of Ophthalmology, Zhongshan Ophthalmic Center, Sun Yat-sen University, Guangzhou, China; [6]Zhejiang University-University of Edinburgh Institute (ZJU-UoE Institute), Zhejiang University School of Medicine, International Campus, Zhejiang University, Haining, China; [7]Reproductive & Developmental Biology Laboratory, National Institute of Environmental Health Sciences (NIEHS), Research Triangle Park, United States; [8]Department of Medicine, University of Florida College of Medicine, Gainesville, United States

**Abstract** Lung cancer with loss-of-function of the LKB1 tumor suppressor is a common aggressive subgroup with no effective therapies. LKB1-deficiency induces constitutive activation of cAMP/CREB-mediated transcription by a family of three CREB-regulated transcription coactivators (CRTC1-3). However, the significance and mechanism of CRTC activation in promoting the aggressive phenotype of LKB1-null cancer remain poorly characterized. Here, we observed overlapping CRTC expression patterns and mild growth phenotypes of individual CRTC-knockouts in lung cancer, suggesting functional redundancy of CRTC1-3. We consequently designed a dominant-negative mutant (dnCRTC) to block all three CRTCs to bind and co-activate CREB. Expression of dnCRTC efficiently inhibited the aberrantly activated cAMP/CREB-mediated oncogenic transcriptional program induced by LKB1-deficiency, and specifically blocked the growth of human and murine LKB1-inactivated lung cancer. Collectively, this study provides direct proof for an essential role of the CRTC-CREB activation in promoting the malignant phenotypes of LKB1-null lung cancer and proposes the CRTC-CREB interaction interface as a novel therapeutic target.

*For correspondence:
lzwu@ufl.edu

Competing interests: The authors declare that no competing interests exist.

## Introduction

Lung cancer is the leading cause of cancer deaths in both men and women in the United States and worldwide (*Siegel et al., 2020*; *Bray et al., 2018*; *Torre et al., 2016*). Global cancer statistics estimated 1,761,007 deaths due to lung cancer in 2018, contributing to about 20% of all cancer deaths (*Bray et al., 2018*). In 2020, there were an estimated 228,820 newly diagnosed lung cancer cases and 135,720 lung cancer deaths in the United States alone (*Siegel et al., 2020*). Non-small cell lung cancer (NSCLC) accounts for approximately 85% of lung cancer cases and includes the major subtypes: lung adenocarcinoma, squamous cell carcinoma, and large cell carcinoma (*Herbst et al., 2018*). While small-molecule inhibitors targeted at driver gain-of-function gene mutations have achieved improved clinical outcomes over conventional cytotoxic therapy, they are currently

available for only a subset of patients with lung cancer harboring specific mutations such as EGFR and ALK mutations (*Howlader et al., 2020*; *Koivunen et al., 2008*; *Paez et al., 2004*). Cancer immunotherapy has emerged as one of the newest treatment options for NSCLC; however, the recent use of immune checkpoint inhibitors, such as those blocking the PD-1/PD-L1 checkpoint pathway, offer durable tumor responses to only a small population of patients generally with high tumor PD-L1 expression and/or high tumor mutational burden (*Herbst et al., 2018*; *Reck et al., 2016*). Therefore, effective treatments for the majority of lung cancer patients remain lacking.

Comprehensive genomic profiling has revealed the genetic landscape of lung cancer (*Cancer Genome Atlas Research Network, 2014*; *Cancer Genome Atlas Research Network, 2012*; *Ding et al., 2008*), identifying inactivating somatic *STK11* gene mutations as a common event in NSCLC. Somatic *STK11* mutations arise preferentially in lung adenocarcinoma where they have been detected in up to 30% of cases (*Cancer Genome Atlas Research Network, 2014*; *Ding et al., 2008*; *Matsumoto et al., 2007*; *Sanchez-Cespedes et al., 2002*). In addition to gene mutations, *STK11* can be inactivated by epigenetic silencing, post-translational modifications, or alterations in its interacting proteins (*Boudeau et al., 2003*; *Esteller et al., 2000*; *Zheng et al., 2009*). The *STK11* gene encodes a serine-threonine kinase, commonly known as liver kinase B1 (LKB1). Lung cancer with LKB1 deficiency exhibits resistance to chemotherapy, targeted therapeutics and especially to immune checkpoint inhibitors in preclinical models and/or human patients (*Chen et al., 2012*; *Carretero et al., 2010*; *Han et al., 2014*; *Skoulidis et al., 2018*; *Schabath et al., 2016*; *Rizvi et al., 2018*). Therefore, the absence of targeted therapies and the lack of benefits of immune checkpoint inhibitors for this common aggressive lung cancer subtype require an urgent search for new therapeutic strategies.

*STK11* was first identified as the cancer susceptibility locus for familial Peutz-Jeghers syndrome (PJS), which is characterized by mucocutaneous pigmentation and gastrointestinal hamartoma with an increased cancer risk (*Hemminki et al., 1998*; *Giardiello et al., 1987*). Somatic inactivation of LKB1 has now been observed in a variety of human cancers besides lung cancer. Importantly, LKB1 loss has been shown to promote cancer progression and increase metastatic potential in the genetically engineered mouse models of lung cancer, melanoma, pancreatic cancer, and endometrial cancer (*Hermanova et al., 2020*; *Ji et al., 2007*; *Liu et al., 2012*; *Peña et al., 2015*). Also, *STK11* mutations are associated with the suppressive immune milieu of the lung tumor microenvironment (*Schabath et al., 2016*; *Koyama et al., 2016*). Thus, *STK11* is a bona fide tumor suppressor gene. A better understanding of the pathogenic downstream signaling induced by LKB1 inactivation will facilitate the identification of rational therapeutic approaches.

The LKB1 kinase is essential for the activation of 14 downstream AMPK family members, such as AMP-activated protein kinase (AMPK) and salt-inducible kinases (SIKs) (*Alessi et al., 2006*; *Shackelford and Shaw, 2009*). Therefore, LKB1 regulates multiple signaling pathways through its various substrates and plays critical roles in regulating cell polarity, metabolism, and growth (*Alessi et al., 2006*; *Shackelford and Shaw, 2009*). Consequently, LKB1 inactivation has the potential to promote tumorigenesis by deregulating downstream cell signaling, such as the defective LKB1-AMPK-mediated energy stress response which has been the focus of many studies (*Shackelford and Shaw, 2009*). However, unlike loss of LKB1, loss of AMPK was found to reduce the growth of murine oncogenic Kras G12D-driven lung cancer (*Eichner et al., 2019*), indicating that AMPK does not mediate LKB1's tumor suppression function. To identify key signaling pathway(s) impacted by LKB1 inactivation in lung cancer, we previously performed an unbiased global gene expression profiling analysis and discovered that multiple cAMP/CREB-regulated targets, such as *LINC00473*, *INSL4*, *NR4A1-3*, and *PTGS2*, were highly expressed in human LKB1-null lung cancer cell lines and primary tumors (*Yang et al., 2019*; *Chen et al., 2016*). The induction of these cAMP/CREB-mediated targets was linked with aberrant hyper-activation of the CRTC (*C*REB-*r*egulated *t*ranscription *co*-activator) family in the context of LKB1 deficiency (*Yang et al., 2019*; *Chen et al., 2016*). In addition, we previously generated an LKB1-null gene signature from 53 lung cancer cell lines to screen the Broad Institute Connectivity Map (CMAP) drug response database and the top 17 compounds that positively correlated with the LKB1-null gene signature were all compounds directly associated with CRTC activation (*Cao et al., 2015*). The CRTC family consists of three members, *CRTC1*, *CRTC2,* and *CRTC3*, which play important roles in metabolism, aging, and cancer (*Altarejos and Montminy, 2011*; *Iourgenko et al., 2003*; *Conkright et al., 2003*; *Tonon et al., 2003*). These three CRTC proteins function as latent transcriptional co-activators and are normally

sequestered in the cytoplasm. In response to cAMP and/or calcium signals, the family of three salt-inducible kinases (SIK1, 2, 3) are inactivated and/or phosphatases become activated, leading to CRTC dephosphorylation. Dephosphorylated CRTCs subsequently translocate to the nucleus and interact with the transcription factor CREB, activating CRE (cAMP-responsive element)-containing promoters. Since SIKs are dependent on LKB1 for its kinase activity, LKB1 deficiency impairs SIKs to phosphorylate CRTCs and consequently leads to an elevated level of unphosphorylated CRTCs, resulting in CRTC nuclear translocation and activation of CREB-mediated transcription. Therefore, the aberrant activation of the SIK-CRTC-CREB signaling axis may serve as a core driver event that underlies the aggressive phenotypes of LKB1-inactivated lung malignancies. This notion is further supported by recent CRISPR/Cas9-mediated gene editing studies revealing that knock-outs of SIK1 and SIK3, but not of other AMPK family members, increased tumor growth in a mouse model of oncogenic KRAS-driven lung adenocarcinoma (*Hollstein et al., 2019*; *Murray et al., 2019*). Therefore, SIKs mediate the major tumor suppressive effects of LKB1 in NSCLC. Moreover, CRTC2 was reported to promote tumor growth in LKB1-deficient NSCLC (*Rodón et al., 2019*). However, the relative contributions of the three CRTC co-activators were not yet defined. Importantly, the role of the aberrant CRTC-CREB activation in LKB1-inactivated lung cancer and its underlying molecular mechanisms remained to be characterized.

In this study, we evaluated the significance and mechanisms of CRTC co-activators in lung cancers using CRISPR/Cas9-mediated knockouts of individual CRTCs and a pan-CRTC inhibitor that blocks all three CRTC co-activators' ability to interact with the CREB transcription factor. Our in vitro and in vivo data provide direct evidence that CRTC activation plays an essential role in the growth of LKB1-deficient lung cancer cells and revealed that targeting gain-of-function CREB activation by interfering with the CRTC-CREB interaction is a potential effective strategy in treating LKB1-inactivated lung cancers.

## Results

### Three CRTC co-activators were expressed at varying levels in lung cancer cells

To assess the individual functional contributions of the three CRTC family members in lung malignancies, we first evaluated their expression patterns by quantitative RT-PCR (RT-qPCR) and western blot analyses of SV40-transformed, non-tumorigenic human bronchial epithelial cells (BEAS-2B), 7 LKB1-wt and 6 LKB1-null NSCLC cell lines. We observed that the three CRTC genes were all expressed at varying levels, with *CRTC2* and *CRTC3* having higher relative expression than *CRTC1* at the transcript level (*Figure 1A*). Also, there were variable CRTC protein levels in all the cell lines examined and protein levels did not tightly correlate with their RNA transcript levels (*Figure 1B*), suggesting potential post-transcriptional regulation. CRTC1 and CRTC2 exhibited predominantly faster migrating bands in all six LKB1-null cancer cells, consistent with enrichment of dephosphorylated forms in the setting of LKB1 deficiency. However, the mobility of CRTC3 appeared relatively unchanged in 4/6 LKB1-null cancer cell lines, suggesting a distinct pattern of post-translational regulation. Therefore, we detected variable levels of expression of all three CRTC genes in immortalized human lung epithelial cells and lung cancer cells, suggesting the potential for both functional redundancies and unique properties. To confirm that LKB1 loss results in the de-phosphorylation and nuclear translocation of CRTCs, two essential steps for CRTC transcriptional activation of CREB target genes (*Iourgenko et al., 2003*; *Conkright et al., 2003*), we determined the phosphorylation status and subcellular localization of the three CRTC proteins by performing phosphatase treatment and subcellular fractionation followed by western blot analysis. As shown in *Figure 1—figure supplement 1A*, endogenous CRTC1, CRTC2, and CRTC3 proteins showed slow migration patterns in LKB1-expressing lung cancer cells (H322). Upon treatment of phosphatase, the mobility of the endogenous CRTC proteins in LKB1-expressing lung cancer cells was shifted to the underphosphorylated forms, which matched the mobility patterns of CRTCs in the LKB1-null lung cancer cells. These data demonstrate that endogenous CRTCs are predominantly phosphorylated in LKB1-expressing cells and dephosphorylated in LKB1-null cells. Further immunoblot analysis of the nuclear and cytoplasmic fractions revealed that the CRTC proteins were predominantly detected as dephosphorylated, nuclear forms in LKB1-null cells and phosphorylated, cytoplasmic forms in LKB1-expressing cells

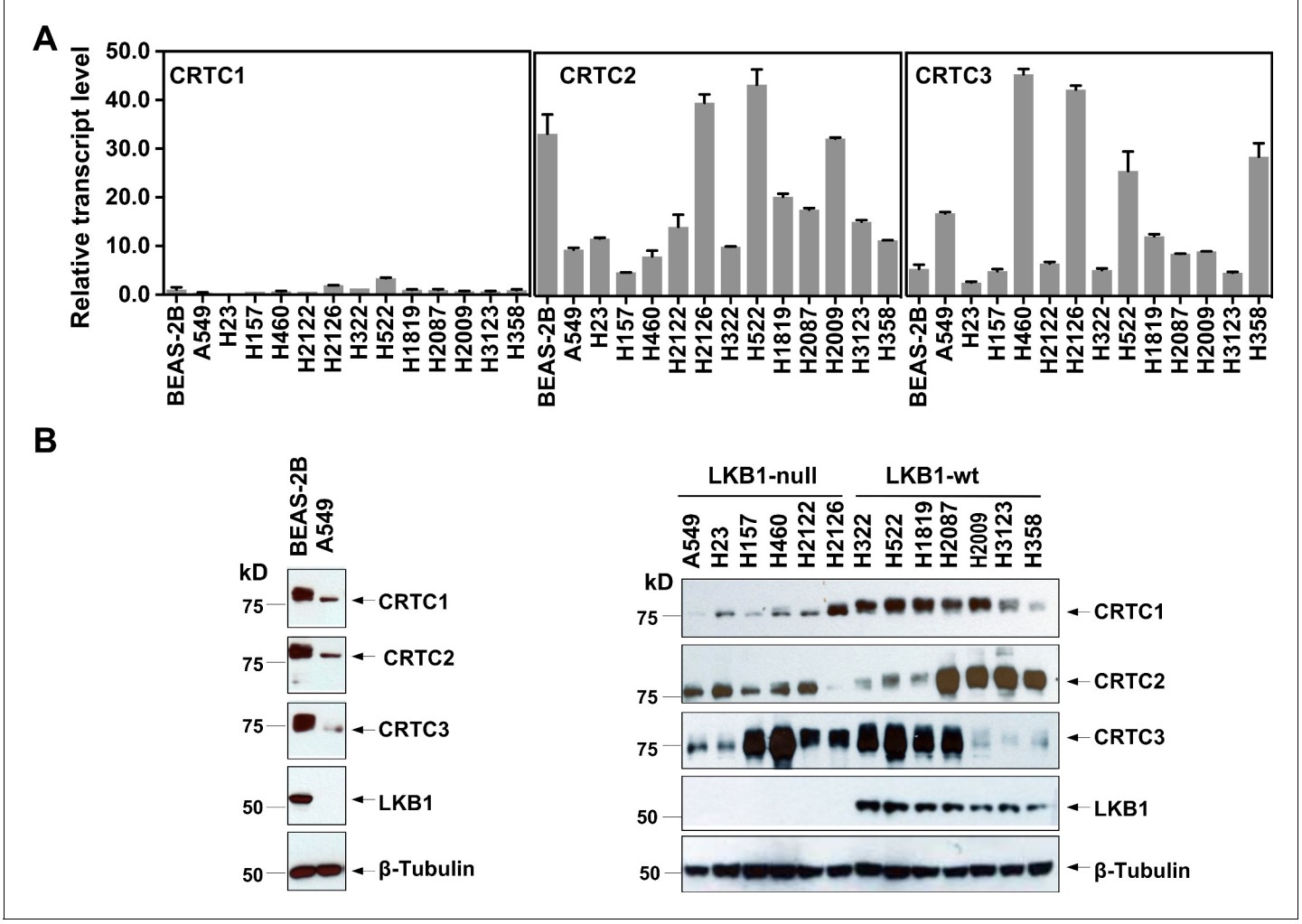

**Figure 1.** Three members of the CRTC co-activator family, *CRTC1*, *CRTC2,* and *CRTC3*, are expressed at varying levels in human lung epithelial and cancer cell lines. (**A**) The transcript levels of the three CRTC genes were determined by RT-qPCR assays. All three CRTC transcript levels were normalized against the level of the housekeeping gene *GAPDH* individually. The expression level of *CRTC1* in BEAS-2B was then assigned as 1, and the expression levels for the three CRTCs in various cell lines were presented as relative values to that of *CRTC1* in BEAS-2B cells. (**B**) The protein levels of three CRTCs and LKB1 were detected by western blotting. Blotting with anti-β-Tubulin was used as a loading control.

The online version of this article includes the following source data and figure supplement(s) for figure 1:

**Source data 1.** Numerical data for A.

**Source data 2.** Unedited immunoblots in B.

**Figure supplement 1.** CRTC1, CRTC2, and CRTC3 showed predominantly de-phosphorylated, nuclear forms in LKB1-null cells (A549), and phosphorylated, cytoplasmic forms in LKB1-expressing cells (H322).

**Figure supplement 1—source data 1.** Unedited immunoblots in A, B.

**Figure supplement 2.** LKB1 re-introduction to LKB1-null lung cancer cells (A549) induced the phosphorylation and cytoplasmic retention of CRTCs.

**Figure supplement 2—source data 1.** Unedited immunoblots in A, B.

**Figure supplement 3.** CRISPR/Cas9-mediated LKB1 knockout in LKB1-expressing lung cancer cells (H322) resulted in the dephosphorylation and nuclear translocation of CRTCs.

**Figure supplement 3—source data 1.** Unedited immunoblots in A, B.

(*Figure 1—figure supplement 1B*). Reintroduction of LKB1 to LKB1-null A549 cells led to an increase in the levels of phosphorylated CRTCs, which correlated with a decrease in nuclear CRTCs and an increase in cytoplasmic CRTCs (*Figure 1—figure supplement 2*). Further, LKB1 knockout in LKB1-expressing H322 cells caused an increase in dephosphorylated, nuclear forms of CRTCs and a decrease in phosphorylated, cytoplasmic forms of CRTCs (*Figure 1—figure supplement 3*). All these data validate LKB1 regulation of CRTC phosphorylation and subcellular localization, further

supporting the existing model where LKB1-dependent SIKs mediate phosphorylation and cyto-plasmic retention of CRTCs. Since all three dephosphorylated, nuclear CRTCs can be detected in LKB1-deficient cells that are capable of co-activating CREB-mediated transcription, we would need to inactivate each gene individually or all three together to determine the contribution of the CRTC co-activation to the transcriptional program in lung cancer cells with LKB1 deficiency.

## CRISPR/Cas9-mediated knock-outs of individual CRTCs reduced expression of the CREB target genes and caused mild effects on NSCLC cell growth

To assess the importance of each CRTC family member in regulating lung cancer cell phenotype, we generated and characterized cells with individual CRTC knockouts. Specifically, human LKB1-null lung cancer A549 cells were transduced with lentiviruses expressing single-guide RNAs (sgRNAs) for each CRTC gene or control sgRNA together with Cas9. Two independent, single knockout clones for each CRTC gene were then selected and CRISPR/Cas9-edited alleles with indels were further val-idated by genomic DNA sequencing (*Figure 2—figure supplement 1*). We observed complete abla-tion of endogenous CRTC proteins in their respective knockout cells, as compared to the parental and control knockout cells by western blotting (*Figure 2A*). Upregulated CRTC1 protein levels were observed in response to CRTC2 knockout or CRTC3 knockout, indicating potential functional com-pensation. These individual CRTC knockout cells showed a reduction in expression of several CREB-mediated target genes, such as *PDE4D*, *INSL4*, *LINC00473*, and *NR4A2*, but not to the extent of their endogenous levels in LKB1-wt lung cancer H522 cells, as assayed by western blotting or RT-qPCR assays (*Figure 2A,B*). Individual CRTC knockout or control A549 cells were further assayed for cellular phenotypes, including cell viability, apoptosis, and anchorage-independent growth by trypan blue exclusion, annexin V/propidium iodide (PI) staining, and soft agar colony formation assays, respectively. We observed that knockout of each individual CRTC gene had only a mild effect on the numbers of viable cells, apoptotic cells, and colonies grown in soft agar (*Figure 2C,D,E*). These data indicate that the CRTC family members may be functionally redundant in regulating lung cancer cell proliferation, survival, and anchorage-independent growth.

## The dnCRTC (CRTC1 CBD-nls-GFP) functioned as a pan-inhibitor for the CRTC-CREB interaction and suppressed the CRTC-CREB signaling axis

Due to the potential functional redundancy of three CRTC coactivators in maintaining malignant cell behaviors of LKB1-null lung cancers, an approach of inhibiting all three CRTCs is required to assess the general role of aberrant CRTC activation in promoting tumorigenesis in LKB1-null lung cancer. The CRTC co-activators contain a highly conserved N terminal CREB-binding domain (CBD) that is responsible for interacting with the transcription factor CREB, and a C terminal transcriptional activa-tion domain (TAD) that is essential for transcriptional activation (*Altarejos and Montminy, 2011*; *Figure 3A*). We, therefore, established a dominant negative approach of blocking the functions of all three CRTC co-activators by competing with endogenous CRTCs for CREB binding. Specifically, we generated a retroviral pMSCV-based dominant negative CRTC (dnCRTC) construct that expresses the CRTC1-CBD-nls-GFP chimeric protein, which contains the CBD of CRTC1 (1–55 aa) fol-lowed by a nuclear localization signal (nls, 'PKKKRKV') and EGFP. This CRTC1-CBD-nls-GFP protein was predicted to bind to CREB but lacks transcriptional activation, consequently interfering with the functions of endogenous CRTC co-activators through competitive CREB binding (*Figure 3A*). We infected human LKB1-null lung cancer A549 cells with the CRTC1-CBD-nls-GFP or GFP (control) ret-roviruses and observed that CRTC1-CBD-nls-GFP was predominantly localized in the nuclear com-partment, while the control GFP showed diffuse cytoplasmic and nuclear signals (*Figure 3B*). The CRTC1-CBD-nls-GFP chimeric protein showed an expected size of ~33 kDa (*Figure 3C*) and sup-pressed the ability of the three CRTC co-activators to activate the CREB-dependent transcription in cAMP response element (CRE)-containing promoter luciferase reporter assays (*Figure 3D*). There-fore, the CRTC1-CBD-nls-GFP chimeric protein functions as a dominant negative mutant for CRTC (dnCRTC), capable of blocking all three CRTCs to co-activate CREB-mediated transcription.

We next determined whether this dnCRTC interacts with the transcription factor CREB on endog-enous CRE-containing gene promoters. Using chromatins prepared from dnCRTC- and control GFP-expressing A549 cells after cross-linking, we performed chromatin immunoprecipitation (ChIP) assay

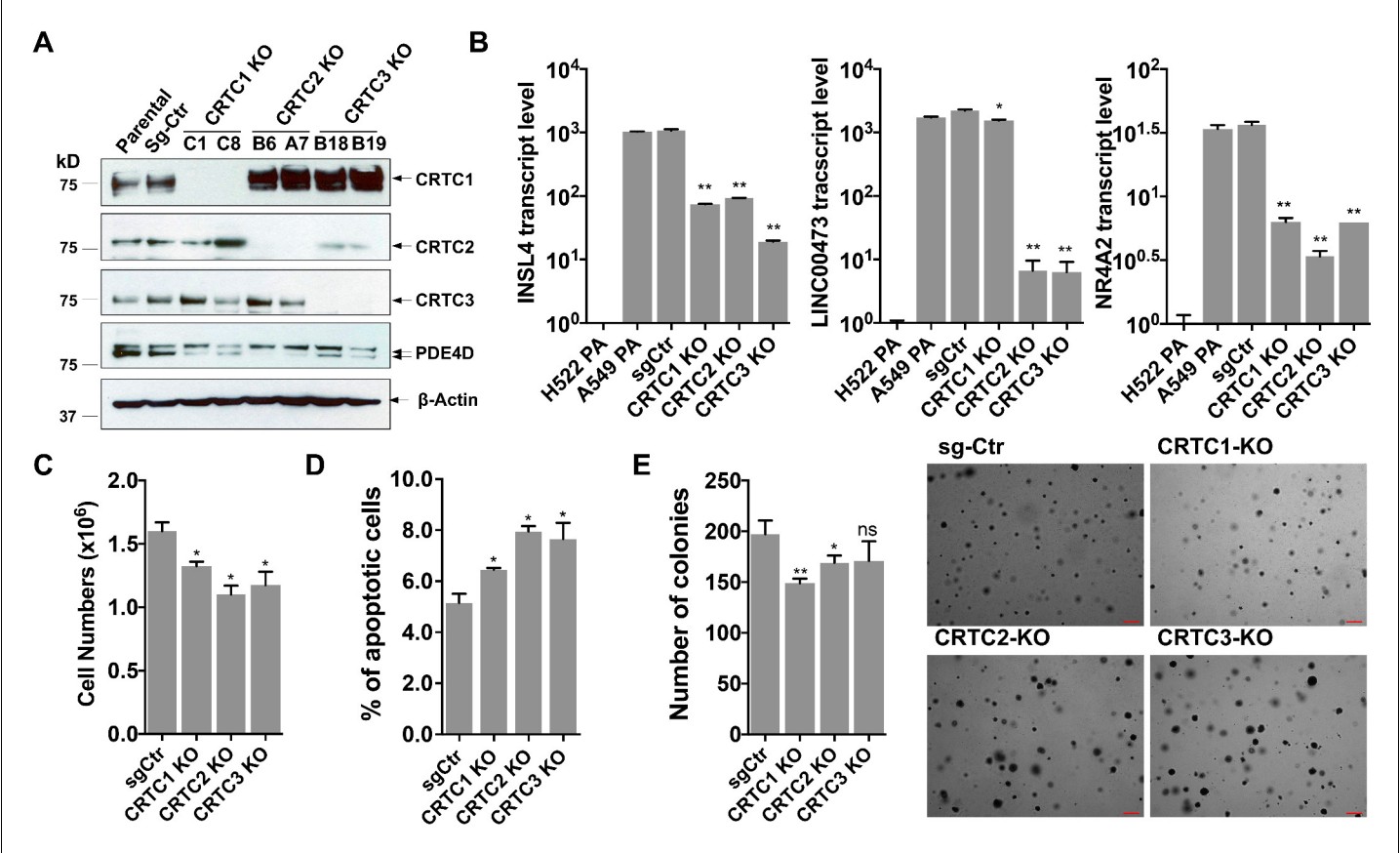

**Figure 2.** Individual knockouts of the CRTC family members in human LKB1-null lung cancer cells inhibit the CREB-mediated target gene expression and moderately affect cell viability and anchorage-independent growth. (A) Western blot analysis of endogenous CRTC proteins in parental A549 cells, A549 cells stably transduced with non-targeting sgRNA, and two independent single knockout clones for each CRTC1, CRTC2, or CRTC3. The protein level of a CREB target gene, PDE4D was also detected. Blotting with anti-β-ACTIN was used as a loading control. (B) The transcript levels of CREB-mediated target genes (INSL4, LINC00473 and NR4A2) were determined by RT-qPCR assays (n = 2). The LKB1-wt cells, H522 parental (PA) cells, were also analyzed. (C,D) Individual CRTC knockout or control cells were cultured at $3 \times 10^5$ cells/well in the 6-well plates for 96 hr. The viable cells were quantified by trypan blue exclusion assay (C), and the number of apoptotic cells was determined by staining with annexin V/propidium iodide (PI) followed by flow cytometry (D). (E) Control and CRTC knockout cells were cultured in soft agar for 14 days, and the resulting colonies were stained by crystal violet and photographed under microscope. The number of colonies was counted using ImageJ. Assays were performed in triplicate. One-way ANOVA test was used to calculate the p values (*p<0.05, **p<0.01, ns p>0.05).

The online version of this article includes the following source data and figure supplement(s) for figure 2:

**Source data 1.** Unedited immunoblots in A.
**Source data 2.** Numerical data for B, C, D, E.
**Figure supplement 1.** CRISPR/Cas9-edited alleles in two independent single knockout clones for each CRTC gene were validated by genomic DNA sequencing.

of dnCRTC or GFP using GFP-trap that consists of anti-GFP V$_H$H nanobodies coupled to agarose beads (ChromoTek) or uncoupled agarose beads as negative control. Western blotting detected CREB in the dnCRTC-ChIP complex, but not in the control GFP ChIP complex (*Figure 3E*), demonstrating a physical association of dnCRTC and CREB. We also observed that the DNA sequences spanning the CRE regions within the promoters of *LINC00473* and *NR4A2*, two genes known to be upregulated by CRTC-CREB activation due to LKB1 deficiency, were significantly enriched in the dnCRTC ChIP complex, but not in the control GFP ChIP complex by RT-qPCR assays (*Figure 3F*). Moreover, ChIP analysis using three CRTC antibodies showed that the enrichment of CRTCs on CRE-containing promoters (*LINC00473* and *NR4A2)* was significantly reduced (*Figure 3G*). Taken together, these data demonstrate that the dnCRTC (CRTC1-CBD-nls-GFP) mutant physically associates with CREB on the CRE-containing gene promoters and blocks the recruitment of endogenous

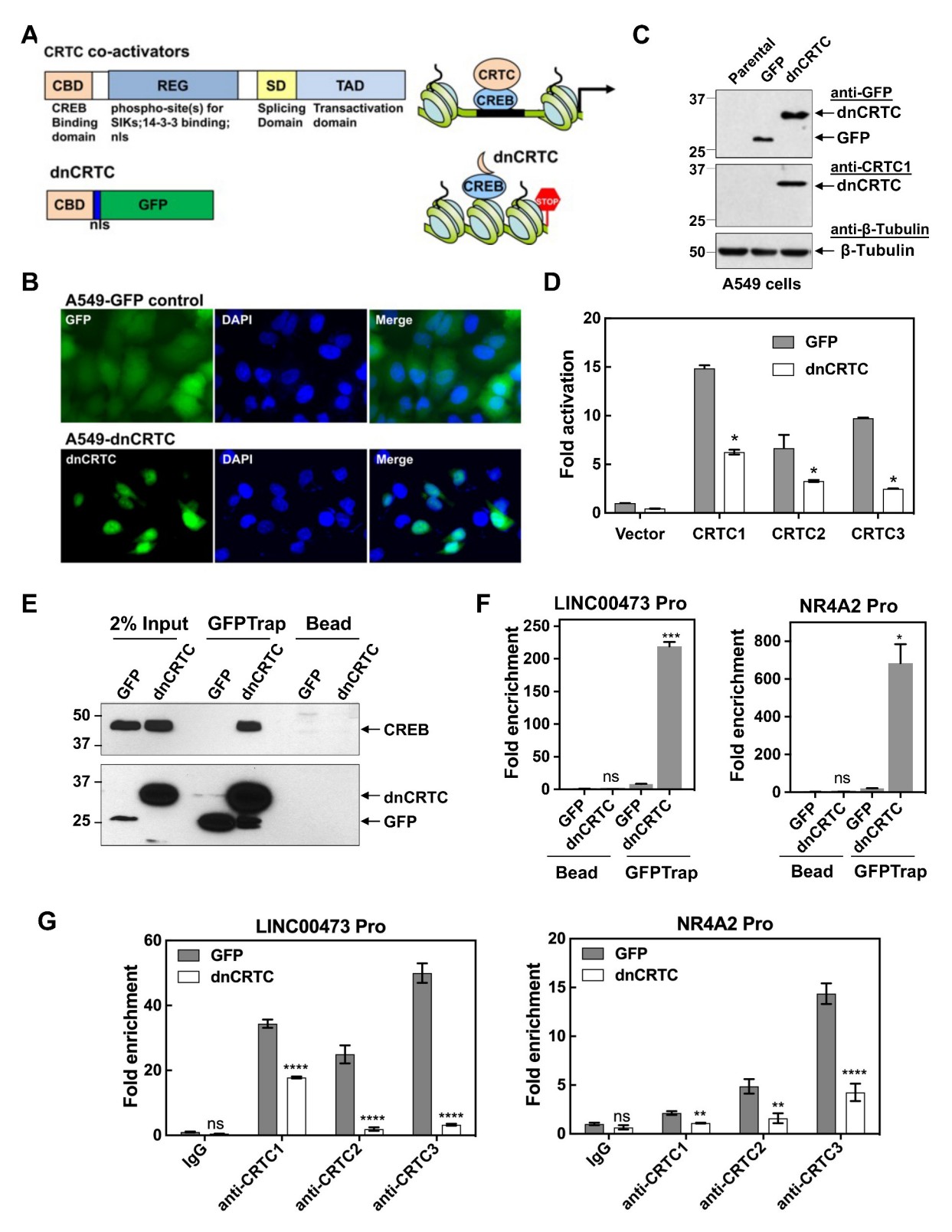

**Figure 3.** A dominant negative CRTC mutant (dnCRTC) interacted with CREB on the target gene promoters and blocked CRTC co-activation of CREB transcription. (A) A diagram of CRTC co-activator and dnCRTC was shown. The dnCRTC consists of CRTC1 (1-55aa) followed by a nuclear localization signal (nls) and GFP, cloned into the retroviral pMSCV vector. (B) A549 cells transduced with pMSCV-dnCRTC retroviruses showed that dnCRTC was predominantly localized in the nuclear compartment (lower), while A549 control cells transduced with pMSCV-GFP retroviruses showed both

*Figure 3 continued on next page*

*Figure 3 continued*

cytoplasmic and nuclear GFP signals (upper). DAPI stained for the nuclei. (**C**) Western blotting validated the expression of dnCRTC in transduced A549 cells. (**D**) Expression of dnCRTC blocked the abilities of CRTC1-3 to activate the pCRE-luc reporter in 293 T cells (n = 2). (**E**) dnCRTC interacts with CREB in the chromatin complex. Cells (GFP-expressing control and dnCRTC-GFP expressing cells) were crosslinked and chromatins were sonicated. GFP-Trap_A (anti-GFP V$_H$H nano body coupled to agarose beads) were used for immunoprecipitation of dnCRTC-GFP proteins which were then blotted with anti-CREB and anti-GFP antibodies. Uncoupled agarose beads were used as control. (**F**) dnCRTC was enriched on the CRE regions of the LINC00473 and NR4A2 promoters. (**G**) dnCRTC reduced the enrichment of endogenous CRTC1, CRTC2, and CRTC3 proteins on the CRE regions of the LINC00473 and NR4A2 promoters. Two-tailed student's t-test was used to calculate the p values (*p<0.05, **p<0.01, ns p>0.05).

The online version of this article includes the following source data for figure 3:

**Source data 1.** Unedited immunoblots in C, E.
**Source data 2.** Numerical data for D, F, G.

CRTC proteins, thus acting as a pan-inhibitor for all three CRTCs in co-activating CREB-mediated transcription.

## Inhibition of CRTC co-activators via dnCRTC effectively blocked the aberrant CREB-mediated transcriptional program in LKB1-null lung cancer cells

To evaluate the extent to which dnCRTC blocks the aberrant CRTC/CREB transcriptional program in LKB1-inactivated lung cancer, we profiled the transcriptomes of dnCRTC vs GFP-expressing cells to identify the affected downstream targets using an unbiased global screen. In brief, LKB1-null A549 lung cancer cells were transduced with dnCRTC and GFP retroviruses for 72 hr, and RNA was then isolated for gene expression profiling using Affymetrix GeneChip Human Transcriptome Array 2.0. Two biological replicates were set up and expression of dnCRTC and control GFP was confirmed by western blotting (*Figure 4A*). Using cut-off criteria of an absolute fold-change >= 2.0 and FDR p<0.05, we identified a total of 274 dnCRTC-regulated differentially expressed genes (dnCRTC-DEGs), including 114 upregulated and 160 downregulated genes (*Supplementary file 1a*); the heat-map and volcano plot were shown in *Figure 4B,C*. Since CRTCs are transcriptional co-activators, we next focused on the top downregulated dnCRTC-DEGs for the validation of the microarray results and confirmed that dnCRTC expression reduced the expression levels of multiple genes by RT-qPCR analysis (*Figure 4D*). These genes include known LKB1 target genes, such as *INSL4, CPS1, NR4A2, LINC00473, NR4A1, PTGS2, SIK1,* and *PDE4D.* We also validated the downregulation of ID1 in dnCRTC-expressing A549 cells (*Figure 4—figure supplement 1*), a CRTC2/CREB target recently reported to be important in regulating LKB1-deficient lung cancer (*Rodón et al., 2019*), although it had a fold change of −1.72 (FDR p<0.05) in dnCRTC-expressing vs. control A549 cells in our profiling experiment. We further showed that the majority of the dnCRTC-regulated gene candidates tested were downregulated in a second LKB1-null cancer cell line H157, but were not affected in 2 LKB1-expressing cell lines (H322 and H522) (*Figure 4—figure supplement 1*). LKB1 knockout in LKB1-expressing cancer cells (H322) led to a significant upregulation of multiple target genes, although not to the extent that was observed in the naturally occurring human LKB1-null lung cancer cells (A549) (*Figure 4—figure supplement 2*). Finally, the expression levels of many dnCRTC-regulated genes were significantly higher in human TCGA lung cancers harboring LKB1 mutations, particularly in lung adenocarcinomas (*Cancer Genome Atlas Research Network, 2014*; *Ding et al., 2008*; *Gao et al., 2013*; *Figure 4—figure supplement 3*). These data indicate that dnCRTCs downregulates expression of multiple genes that are highly expressed in LKB1-null lung cancer.

Ingenuity Pathway analysis revealed that CREB and cAMP are upstream regulators of gene expression changes observed in dnCRTC-expressing vs control A549 cells (*Figure 4E*). Moreover, we analyzed the dnCRTC-DEGs for predicted CRE sites on their promoters (−3 kb to 300 bp from transcription start site) using the CREB Target Gene Database (*Zhang et al., 2005*) and found that 169 of 274 (~61.7%) dnCRTC-DEGs contain predicted or experimentally verified CRE sites, which supports that dnCRTC affects a large set of CREB-regulated transcriptional loci (*Supplementary file 1a*). By incorporating a recently published ChIP-sequencing study that investigated the genome-wide binding profiles of CREB and CRTC2 in LKB1-null A549 cells (*Rodón et al., 2019*), we found that the dnCRTC-DEGs exhibited significant enrichment in CREB and CRTC2 binding around their

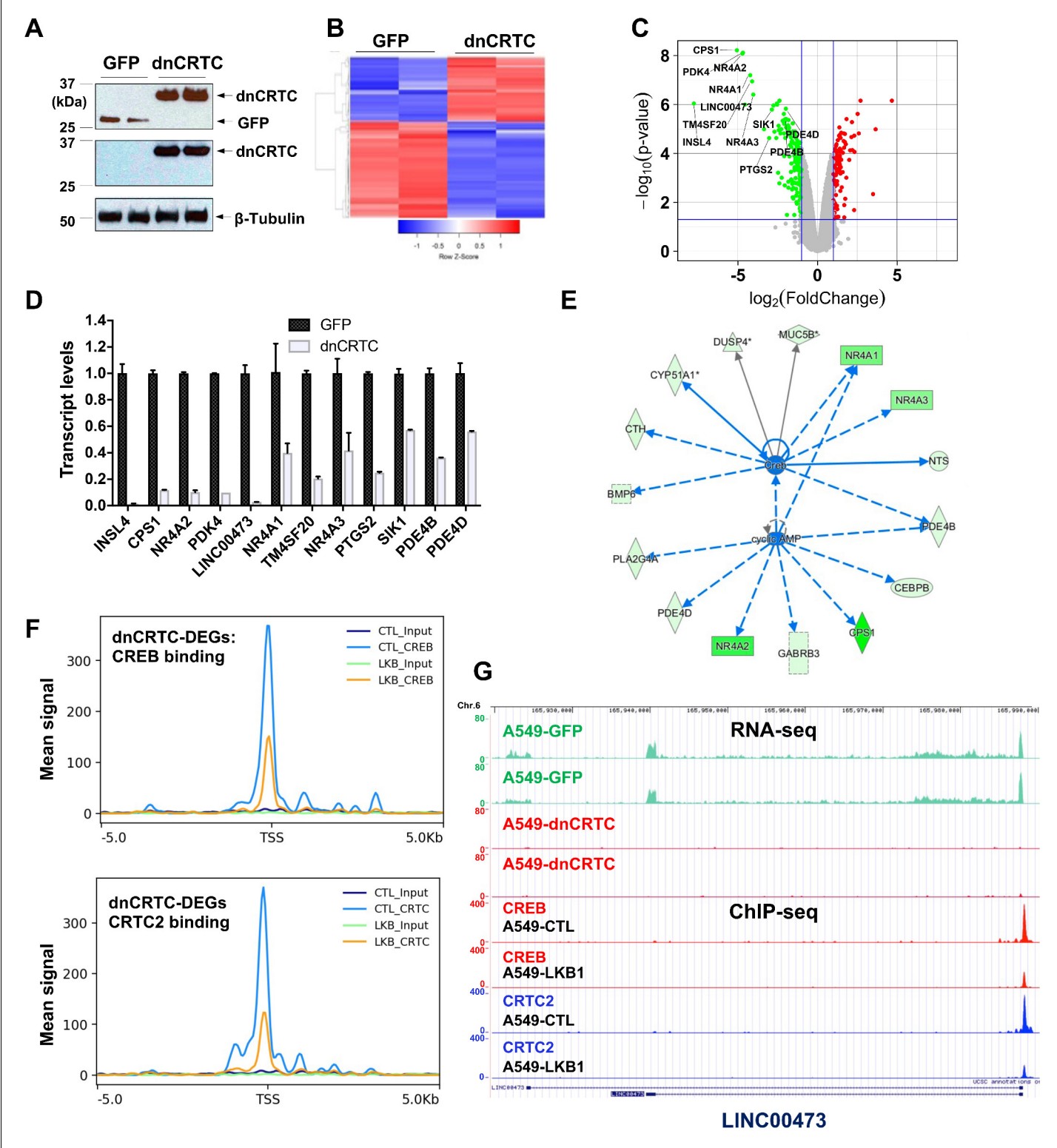

**Figure 4.** Gene expression profiling revealed dnCRTC repressed CRTC-CREB target gene expression. (**A**) Western blotting confirmed dnCRTC-expressing and GFP-expressing control cells. (**B, C**) The heatmap and volcano plots showed gene expression changes in dnCRTC-expressing and GFP-expressing cells. (**D**) The RT-qPCR analysis validated differential expressed genes (DEGs) in dnCRTC-expressing A549 cells. (**E**) IPA analysis identified CREB and cAMP as upstream regulators for gene signature changes due to dnCRTC expression. (**F**) Analysis of CREB and CRTC2 binding of dnCRTC-DEGs in a ChIP-seq dataset. (**G**) CREB and CRTC2 binding peaks were shown in the LINC00473 target gene locus from the ChIP-seq analysis (lower panel). The mapped peaks of sequence reads from RNA-seq of A549-GFP and -dnCRTC cells was also shown (upper panel).

*Figure 4 continued on next page*

*Figure 4 continued*

The online version of this article includes the following source data and figure supplement(s) for figure 4:

**Source data 1.** Unedited immunoblots in A.

**Source data 2.** Numerical data for D.

**Figure supplement 1.** Effects of dnCRTC expression on gene expression in LKB1- expressing and LKB1-null lung cancer cells.

**Figure supplement 1—source data 1.** Numerical data for bar graphs.

**Figure supplement 2.** CRISPR/Cas9-mediated LKB1 knockout in in LKB1-expressing lung cancer cells led to enhanced expression of multiple dnCRTC-regulated targets.

**Figure supplement 2—source data 1.** Numerical data for bar graphs.

**Figure supplement 3.** The Box and whisker plots show gene expression levels in LKB1 mutant (Mut) and wildtype (Wt) groups of lung adenocarcinoma (TCGA-LUAD, PanCancer Atlas) and lung squamous cell carcinoma (TCGA-LSCC, PanCancer Atlas).

transcription start sites (TSS), while the binding of CREB and CRTC2 was reduced upon reintroduction of LKB1 (*Figure 4F*). This analysis revealed 97 of 274 (~35%) dnCRTC-DEGs (60 down-regulated and 37 up-regulated) having CREB-binding and CRTC2-binding peaks within −3 kb to 300 bp from TSS; and 73 of 274 (~27%) dnCRTC-DEGs (45 down-regulated and 28 up-regulated) having both the CREB and CRTC2 binding peaks within −500 bp to 100 bp from TSS (*Supplementary file 1a*). This list includes multiple known CRTC/CREB targets, such as *NR4A2, LINC00473,* and *PTGS2*. A representative close-up view of the CREB and CRTC2 binding on the *LINC00473* gene locus was shown (*Figure 4G*). The mapped peaks of sequence reads from our RNA-seq re-analysis of A549-GFP and -dnCRTC cells were also shown. Overall, we identified a list of direct dnCRTC-regulated genes, which represent an extensive set of the potential critical mediators for CRTC-CREB activation in promoting lung cancer cell growth.

To gain further insights into the biological impact of dnCRTC expression, we performed gene set enrichment analysis (GSEA) of the transcriptomic data from dnCRTC-expressing vs control GFP-expressing A549 control cells using gene sets obtained from the Molecular Signatures Database. Several oncogenic gene signatures, such as Shh-regulated gene set, RB loss/E2F1-regulated gene set, NFE2L2-regulated gene set, PDGF-regulated gene set, KRAS-regulated gene set, were found to be significantly altered with negative enrichment scores (*Supplementary file 1b*), indicating that a majority of genes in these oncogenic gene sets were significantly under-expressed in dnCRTC-expressing cells. Therefore, our dnCRTC mutant serves as a useful tool for blocking the extensive CRTC/CREB transcriptional program and oncogenic signaling. These data also suggest that dnCRTC expression has the potential to negatively impact the malignant behaviors of LKB1-deficient lung cancer cells.

## LKB1-null, but not LKB1-wt, NSCLC cells were sensitive to dnCRTC-induced inhibition of CRTC co-activators in vitro

To determine whether LKB1-null lung cancer cells depend on CRTC-CREB activation for growth and survival, we next assessed the functional impact of blocking the CRTC-CREB interaction via dnCRTC by analyzing the effect on lung cancer cell growth. We first performed competition assays using two LKB1-null (A549 and H157) and two LKB1-wt (H322 and H522) NSCLC cells. These cells were transduced with dnCRTC or control GFP retroviruses at an infection rate of ~40–60%, and then the percentages of GFP-positive cell populations were quantified at 3-day intervals over a total of 24 days starting at day 3 following viral infection. We observed a progressively reduced percentage of LKB1-null cells (A549 and H157) that expressed dnCRTC, while the percent of the GFP-control cells remained stable (*Figure 5A*). In contrast, the percentage of LKB1-positive cells (H322 and H522) was not significantly affected (*Figure 5A*). These results showed that dnCRTC expression has a negative effect on the proliferation of LKB1-null tumor cells, but not of LKB1-positive cells, indicating that CRTC activation is critical for LKB1-null cell growth. We also sorted the GFP +populations from dnCRTC- and GFP-transduced cells and performed functional comparisons. Expression of dnCRTC and GFP was first confirmed by western blotting (*Figure 5B*). We observed that dnCRTC expression induced a significant inhibition of cell growth in LKB1-null cells (A549 and H157), but not in LKB1-positive cells (H322 and H522) (*Figure 5C*). Importantly, dnCRTC expression did not affect cell proliferation in normal lung epithelial cells (BEAS-2B) (*Figure 5—figure supplement 1A-C*). Moreover, colony formation and soft agar colony formation assay showed that dnCRTC expression blocked

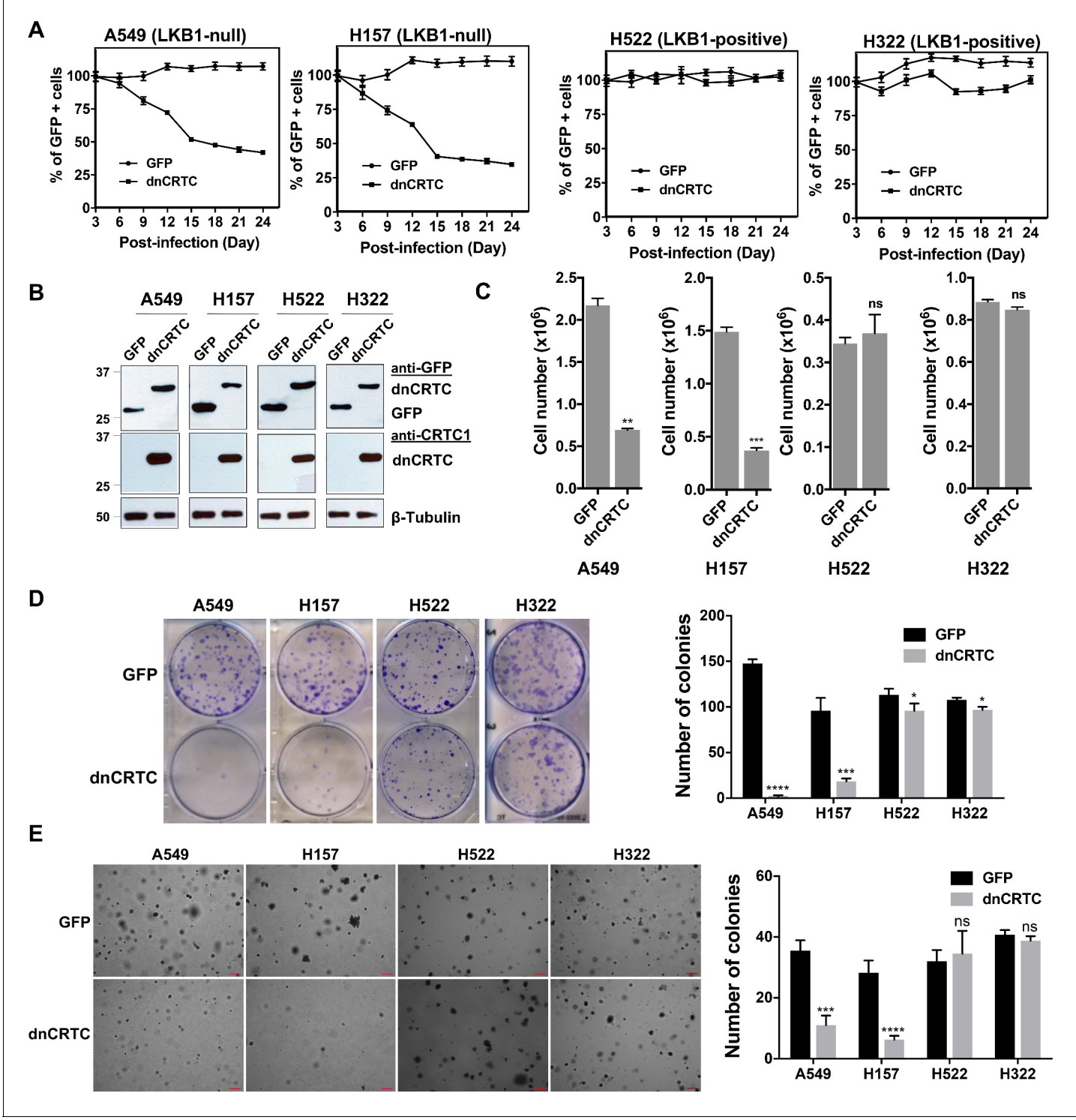

**Figure 5.** dnCRTC expression suppressed the growth of LKB1-null but not LKB1-positive lung cancer cells. (**A**) Two LKB1-null (A549 and H157) and two LKB1-positive (H322 and H522) NSCLC cells were transduced with dnCRTC or control GFP retroviruses. The MOI was optimized to obtain an infection rate of 40–60%, and then the percentage of GFP-positive cells was determined by FACS analysis every 3 days for a total of 24 days starting at day 3 post-infection. The percentage of GFP-positive cells at day three post-infection was considered as 100%, and the remaining data were normalized (n = 3). (**B, C**) The GFP-positive cells for dnCRTC- and GFP-transduced cells were sorted and confirmed for dnCRTC and GFP expression by western blotting (**B**). Sorted cells were also cultured at $2 \times 10^5$ (for H322 and H522) or $3 \times 10^5$ (for A549 and H157) cells/well in the six-well plates for 96 hr and viable cells were counted using trypan blue exclusion test (**C**) (n = 3). (**D**) Transduced cells were cultured at 400 cells/well in six-well plates for 14 days and colonies were stained by crystal violet and photographed. The number of colonies in each well was counted using ImageJ. Assays were performed

*Figure 5 continued on next page*

*Figure 5 continued*

in triplicate. (E) Transduced cells were cultured in soft agar gels and colonies were stained by crystal violets, photographed and counted. The number of colonies from each image was counted using ImageJ. Assays were performed in triplicate. Scale bars, 200µM. Only colonies with a diameter higher than 50 µm were counted (n = 3). Two-tailed student's t-test was used to calculate the p values (*p<0.05, **p<0.01, ***p<0.001, ****p<0.0001, ns p>0.05).

The online version of this article includes the following source data and figure supplement(s) for figure 5:

**Source data 1.** Numerical data for A, C, D, E.
**Source data 2.** Unedited immunoblots in B.
**Figure supplement 1.** Effects of dnCRTC expression on the growth of human immortalized lung bronchial epithelial BEAS-2B cells and mouse LKB1-null NSCLC mLSCC[LP] cells.
**Figure supplement 1—source data 1.** Unedited immunoblots in A, D.
**Figure supplement 1—source data 2.** Numerical data for C, F, G.

colony-forming potential and anchorage-independent growth in LKB1-null cells (A549 and H157), but not in LKB1-positive cells (H322 and H522) (*Figure 5D,E*). Expression of dnCRTC also had a similar negative effect on the growth of a mouse lung squamous carcinoma cell line, which was derived from a mouse model deficient of the tumor suppressors LKB1 and PTEN (*Liu et al., 2019*; *Figure 5— figure supplement 1D-G*). These results demonstrated that LKB1-null NSCLC cells are specifically sensitive to dnCRTC expression; therefore, they are highly dependent on the CRTC-CREB activation for growth.

## Inhibition of CRTC co-activators via dnCRTC expression blocked lung tumor growth and metastatic colonization in vivo

We further determined the effects of dnCRTC expression on the growth and metastatic colonization of lung cancer using subcutaneous and orthotopic NSCLC xenograft models. For subcutaneous xenograft models, luciferase-expressing LKB1-inactivated lung cancer cells (A549-luc and H157-luc) were transduced with retroviruses expressing dnCRTC or control GFP for 72 hr, and then dnCRTC or GFP-transduced cells ($10^6$ cells per mouse) were subcutaneously implanted into immunodeficient NOD/SCID mice. The dnCRTC cohorts had reduced growth of xenograft tumors compared to the GFP control cohorts, as demonstrated by the reduced tumor growth rate, size and weight (*Figure 6A–D,F–I*). Immunohistochemical analysis revealed a decreased number of Ki-67-positive proliferating cells in the dnCRTC-expressing xenograft tumors in comparison with the control GFP group (*Figure 6E,J*). Since the tumor cells used in these xenograft assays were unsorted and not 100% transduced, we performed western blot analysis on the excised xenograft tumors and observed markedly reduced dnCRTC expression when compared to dnCRTC-transduced cells at the time of the injection. In contrast, GFP expression was similar between the excised GFP xenograft tumors and GFP-transduced cells at the time of injection (*Figure 6—figure supplement 1*). These results indicate that the residual small xenograft tumors in the dnCRTC group were likely derived from cells with low or no dnCRTC expression, further supporting the tumor inhibitory effect of dnCRTC expression on the growth of LKB1-null lung cancers.

We also studied the effect of dnCRTC expression on the ability of lung cancer cells to undergo vascular extravasation and lung colonization using orthotopic NSCLC xenograft models. Here, dnCRTC-expressing or control GFP-expressing A549-luc cells or H157-luc cells ($2 \times 10^6$ per mouse) were intravenously injected into immunodeficient NOD/SCID mice and lung tumor burden was monitored. We observed that mice injected with dnCRTC-expressing A549-luc or H157-luc cells, compared to mice with their control GFP counterparts, had reduced tumor burden, a decreased number of surface tumor nodules and smaller tumor areas in the lung, as assessed by bioluminescent imaging (*Figure 7A,D*), fluorescence imaging (*Figure 7B,E*), and H and E staining of lung sections (*Figure 7C,F*). Taken together, these data showed that expression of dnCRTC blocked lung cancer growth and colonization in vivo, indicating that the CRTC-CREB activation is essential for the growth and progression of LKB1-null lung cancer.

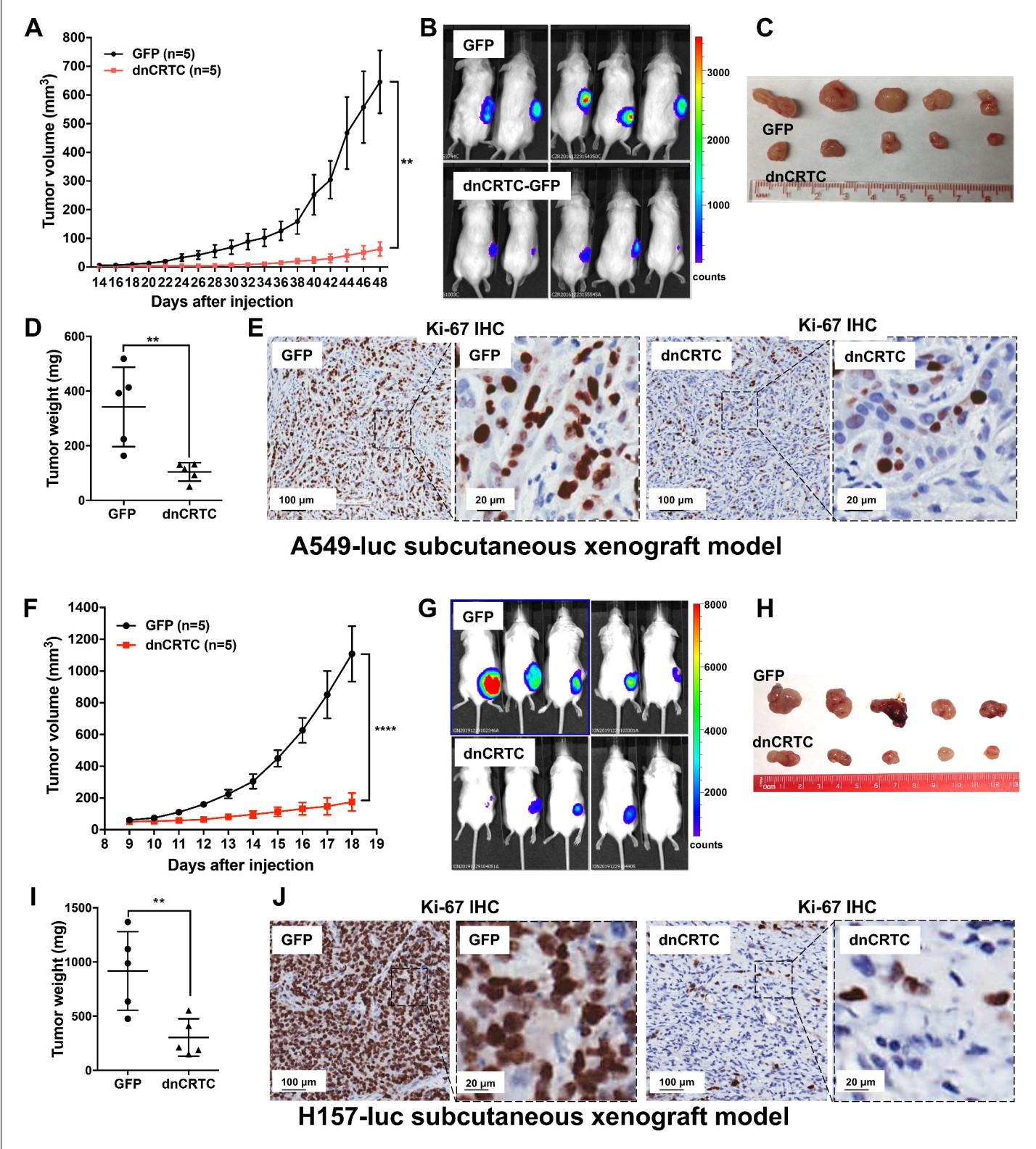

**Figure 6.** Expression of dnCRTC significantly inhibited the growth of LKB1-null NSCLC xenograft tumors. (A–E) A549-luc were transduced with GFP control or dnCRTC for 72 hr and the transduced cells ($1 \times 10^6$ per mouse) were injected subcutaneously to the right flanks of NOD/SCID mice. Tumor volumes of two cohorts (n = 5 each) were measured every two days starting from day 14 until day 48 (A). The bioluminescent images of mice (B), excised tumors (C) and tumor weights (D) as well as Ki-67 immunohistochemical staining of xenograft tumor sections (E) were shown. (F–J) H157-luc

*Figure 6 continued on next page*

*Figure 6 continued*

were transduced with GFP control or dnCRTC for 72 hr and the transduced cells ($1 \times 10^6$ per mouse) were injected subcutaneously to the right flanks of NOD/SCID mice. Tumor volumes of two cohorts (n = 5 each) were measured daily from day 9 to day 18 (**F**). The bioluminescent images of mice (**G**), excised tumors (**H**), tumor weights (**I**) and Ki-67 immunohistochemical staining (**J**) were shown. Scale bars: 100 μm (left panels), 20 μm (right panels). Two-tailed student's t-test was used to calculate the p values (\*\*p<0.01, \*\*\*\*p<0.0001).

The online version of this article includes the following source data and figure supplement(s) for figure 6:

**Source data 1.** Numerical data for A, D, F, I.

**Figure supplement 1.** Expression of GFP and dnCRTC in the transduced human LKB1-null lung cancer cells at the time of tumor cell injection and in the resulting xenograft tumors.

**Figure supplement 1—source data 1.** Unedited immunoblots in A, B.

## Discussion

Lung cancer carrying somatic LKB1 inactivation is a common aggressive molecular subtype with very limited treatment options. Since replacing loss-of-function tumor suppressor mutations is challenging, drug therapeutic efforts have been directed towards identifying and understanding the effector pathways that mediate LKB1 tumor suppression in order to uncover new therapeutic strategies. An important function of LKB1 is its ability to activate SIKs which then phosphorylate and negatively regulate the family of three CREB-regulated transcriptional co-activators (CRTC). We and others have shown that the loss of LKB1 directly leads to CRTC activation and extensive, elevated CRTC1-CREB-mediated transcription in human lung cancer cells and primary tumors (*Yang et al., 2019*; *Chen et al., 2016*; *Cao et al., 2015*; *Hollstein et al., 2019*; *Murray et al., 2019*; *Rodón et al., 2019*). More recently, studies of genetically engineered mouse models of oncogenic *KRAS*-induced lung cancer revealed that SIKs, but not other AMPK family members, mediate the major tumor suppression function of LKB1 (40, 41). These molecular and genetic data support the model that aberrant CRTC-CREB transcriptional activation mediates the major LKB1-null malignancy. However, direct evidence for the importance of CRTC activation in promoting tumorigenesis was lacking. Also, whether there is a specific role for individual CRTC 1–3 family members was unknown. In this study, we showed overlapping expression and the potential for functional redundancy of three CRTC co-activators in lung cancers. Therefore, we designed and validated a pan-CRTC dominant negative inhibitor as a useful tool for blocking all three CRTC co-activator function. Our new mechanistic and functional data demonstrated an essential, general role for CRTC activation in maintaining the malignant phenotypes of LKB1-inactivated lung cancer and identified the CRTC-CREB interaction as a valuable molecular target for development of new therapies for lung cancer with LKB1 deficiency.

The findings in this study further emphasize the importance of CRTC activation in tumorigenesis. We initially identified CRTC1 as a fusion partner with the Notch transcriptional co-activator MAML2, due to a t(11;19) chromosomal translocation in mucoepidermoid carcinoma (MEC), the most common salivary gland malignancies and lung tumors (*Tonon et al., 2003*). This fusion event leads to a chimeric CRTC1-MAML2 protein which is composed of the CREB-binding domain (CBD) of CRTC1 (42aa) fusing to the transcriptional activation domain (TAD) of MAML2 (983aa) (*Tonon et al., 2003*; *Wu et al., 2005*). The CRTC1-MAML2 fusion binds to CREB via the CRTC1 CBD and potently activates CREB-dependent transcription through its MAML2 TAD (*Wu et al., 2005*; *Coxon et al., 2005*; *Chen et al., 2015*), which contribute to the fusion's major oncogenic activity (*Wu et al., 2005*; *Chen et al., 2021*). These data demonstrate a critical role of CRTC activation in MEC tumorigenesis. In our previous studies, we also showed that LKB1-deficiency led to CRTC activation of many CREB-dependent genes, including NR4A2, PTGS2 (aka COX-2), LYPD3, INSL4, and LINC00473, which play important roles in cancer cell growth, survival or invasive properties (*Chen et al., 2016*; *Cao et al., 2015*; *Gu et al., 2012*; *Komiya et al., 2010*). Recently, other groups reported that SIKs were the major AMPK family members that mediate LKB1 tumor suppression and SIK knockouts enhanced CRTC target gene expression (*Hollstein et al., 2019*; *Murray et al., 2019*). Furthermore, CRTC2 downregulation inhibited the growth of LKB1-deficient NSCLC (*Rodón et al., 2019*). All these data support a new model that LKB1-SIK genetic alterations represent a distinct mechanism for the constitutive CRTC-CREB activation that is critical for the tumorigenesis and progression of NSCLC. In this study, we performed expression and functional assays to determine the relative contributions of three CRTC co-activators in lung cancer cells. Our data showed that all three CRTC co-activators

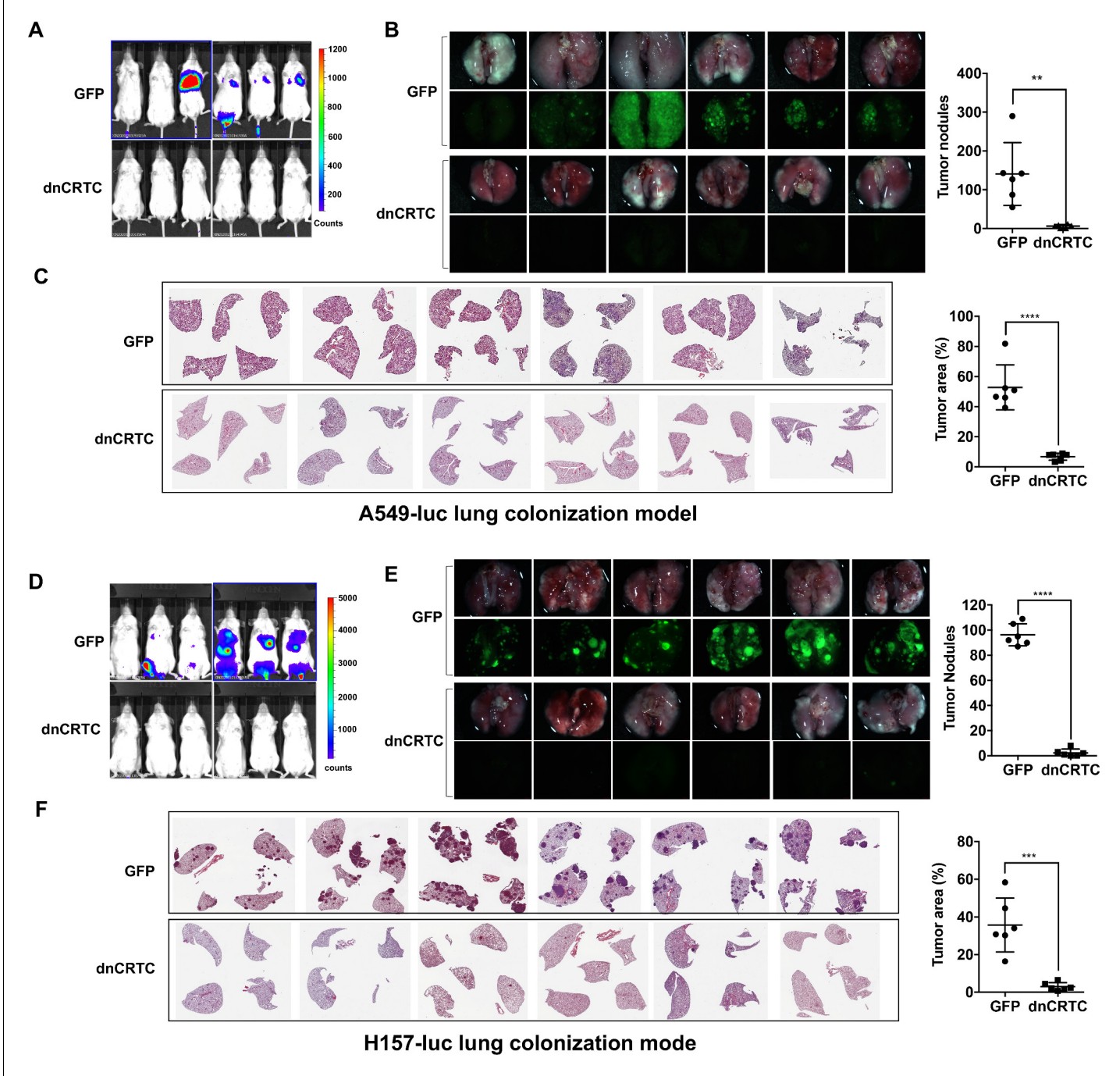

**Figure 7.** Expression of dnCRTC reduced lung colonization of LKB1-null lung cancer cells. (A–C) Luciferase-expressing LKB1-null A549 cells (A549-luc) were transduced with retroviruses expressing GFP control or dnCRTC for 72 hr, and transduced cells ($2 \times 10^6$ cells per mouse) were intravenously injected to NOD/SCID mice (n = 6 each). Eight weeks after injection, lung colonization was assessed by bioluminescent imaging (A). Lungs were dissected and bright field and GFP fluorescence images were shown (B). The number of surface tumor nodules with visible GFP signal per lung of each mouse was quantified and presented (right panel). Representative H and E staining of lung sections were shown (C). Tumor area was calculated from multiple H and E-stained lung sections from each mouse and presented as a percentage of tumor area to total lung area (right panel). (D–F) Luciferase-expressing LKB1-null H157 lung cancer cells (H157-luc) were transduced with retrovirus expressing GFP control or dnCRTC for 72 hr, and transduced cells ($2 \times 10^6$ cells per mouse) were intravenously injected to NOD/SCID mice (n = 6 each). Four weeks after injection, lung colonization was assessed by bioluminescent imaging (D). Lungs were dissected and bright field and GFP fluorescence images were shown (E). The number of tumor nodules with visible GFP signal per lung of each mouse was quantified (right panel). Representative H and E staining images of lung sections were shown (F). Tumor area was calculated from multiple H and E-stained lung sections from each mouse and presented as a percentage of tumor area to total lung area (right panel). The p values were calculated by two-tailed student's t-test (**p<0.01, ***p<0.001, ****p<0.0001).

*Figure 7 continued on next page*

*Figure 7 continued*

The online version of this article includes the following source data for figure 7:

**Source data 1.** Numerical data for B, C, E, F.

(CRTC1-3) are expressed at various levels in lung cancers and that CRISPR/Cas9-mediated knockouts of individual CRTCs only partially reduced the LKB1 target gene expression and had very moderate impact on lung cancer cell proliferation, colony formation, and anchorage-independent growth. It should be noted that a recent study reporting CRTC2 shRNA knockdown or knockout in polyclonal cells impaired soft agar formation but did not affect cell proliferation (*Rodón et al., 2019*). In our study, two CRTC2 KO single clones only showed minimal inhibition of cell proliferation and soft agar colony formation; this discrepancy could be explained by the upregulation of CRTC1 in the CRTC2 KO clones. Therefore, these data indicate the presence for functional redundancy of the three CRTC co-activator family members in driving aberrant CREB transcriptional program and lung cancer malignant phenotypes, thus suggesting that general inhibition of all CRTCs is required for blocking the aberrant CRTC-induced transcriptional program and lung tumorigenesis.

We subsequently developed a dominant negative mutant dnCRTC to block all three CRTC function. This dnCRTC binds to CREB but is defective in transcriptional activation, consequently forming an inactive transcriptional complex with CREB and interfering with the ability of all three CRTCs to co-activate CREB-mediated transcription.Expression of this pan-CRTC inhibitor efficiently and extensively inhibited the aberrantly activated CREB-mediated transcriptional program induced by LKB1 deficiency, including INSL4, CPS1, NR4A1-3, LINC00473, PTGS2, SIK1, PDE4B, and PDE4D. It should be noted that SIK1 is a transcriptional target induced by cAMP-CREB signaling (*Wein et al., 2018*) and it was upregulated in LKB1-null cancer cells (*Chen et al., 2016*). SIK1 downregulation by dnCRTC was only observed in LKB1-null, but not LKB1-wt cancer cells in this study (*Figure 4—figure supplement 1*), which is unlikely to have functional consequences as SIK1 kinase is impaired in LKB1-null cancer cells. Integrated analysis of the dnCRTC-regulated DEGs from our gene expression profiling with the published ChIP-seq data (*Rodón et al., 2019*) revealed the direct target genes downstream of CRTC activation, which include known and potential novel mediators of aberrant CRTC activation in LKB1-inactivated cancer. Future studies of these mediators of CRTC activation and their potential cross-talk with other signaling pathways will enhance our molecular understanding of the loss-of-LKB1 tumor suppression in lung cancer. Since dnCRTC acts as a pan-CRTC inhibitor, it has the potential to serve as an invaluable research tool for dissecting the role of deregulated CRTC activation in various disease settings, such as cancers with aberrant CRTC activation (e.g. LKB1 deficiency, the CRTC1-MAML2 fusion), diabetes with CRTC activation that contributes to high blood glucose levels as well as neurological conditions such as depression and memory.

In this study, we utilized this pan-CRTC inhibitor to probe the functional impact of blocking the CRTC-CREB activation on the growth of multiple NSCLC cell lines and xenograft models. We showed that dnCRTC expression caused significant growth inhibition in LKB1-null, but not LKB1-wt cancer cells and normal lung epithelial cells. The growth and lung colonization of LKB1-null lung cancer cells were specifically susceptible to inhibition of CRTC coactivators. These results demonstrate an essential role of aberrant CRTC activation in supporting the malignant phenotypes of LKB1-inactivated lung cancers. This current study focused on addressing the impact of dnCRTC expression on cancer cells using human lung cancer cell lines and xenograft tumors in immunocompromised mice. Since LKB1 loss in lung cancer are associated with immunosuppressive tumor microenvironment and the resistance to immune checkpoint blockade in preclinical models and clinical trials (*Skoulidis et al., 2018*; *Koyama et al., 2016*; *Kitajima et al., 2019*), future research examining the effect of CRTC inhibition in lung cancer genetic mouse models is needed to elucidate the role of CRTC co-activators in lung cancer progression, tumor microenvironment, and therapeutic responses.

As the malignant phenotype of LKB1-inactivated lung cancers are specifically dependent on aberrant CRTC co-activation of the oncogenic CREB transcriptional program, targeting the CRTC-CREB interaction, hence, the active CRTC-CREB transcription complex, may selectively inhibit LKB1-deficient tumors with minimal effects on normal cells as demonstrated by our data. The strategy of blocking the assembly of active CRTC-CREB transcriptional complex and subsequently inhibiting extensive CRTC target genes has the advantage of simultaneously inhibiting multiple deleterious cell

signals which predicts a greater challenge for resistant cancer cell clones to emerge. We propose that inhibition of the CRTC-CREB interaction should reverse the oncogenic activity of CRTC activation. For instance, peptides and peptide-like molecules designed to recapitulate a critical interaction motif will have the potential in selectively targeting the CRTC/CREB interaction interface and consequently inhibit lung cancer growth. The crystal structural analysis has provided important molecular information of the core CRTC/CREB transcriptional complex (*Luo et al., 2012*; *Song et al., 2018*), revealing that CRTC CBD interacts with CREB basic leucine zipper (bZIP) domain forming a 2:2 complex on CRE-containing DNA. CRTC interacts with both CREB and DNA through highly conserved residues are crucial for the complex assembly and CREB stabilization on DNA. With insights from the crystal structural studies and further understanding of the assembly and composition of the CRTC/CREB transcriptional complex, new approaches can be developed to inhibit the oncogenic CRTC/CREB transcriptional program and block the progression of lung cancers.

Collectively, our study provides direct proof for a critical role of the CRTC-CREB activation in maintaining the malignant phenotypes of LKB1-inactive lung cancers and reveals direct inhibition of the CRTC-CREB transcriptional complex via targeting the CRTC-CREB interface as a novel, promising therapeutic approach.

# Materials and methods

**Key resources table**

| Reagent type (species) or resource | Designation | Source or reference | Identifiers | Additional information |
|---|---|---|---|---|
| Gene (*Homo sapiens*) | STK11 | GenBank | Gene ID: 6794 | This gene is commonly known as LKB1 in the field |
| Gene (*Homo sapiens*) | CRTC1 | GenBank | Gene ID: 23373 | |
| Gene (*Homo sapiens*) | CRTC2 | GenBank | Gene ID: 200186 | |
| Gene (*Homo sapiens*) | CRTC3 | GenBank | Gene ID: 64784 | |
| Gene (*Homo sapiens*) | NR4A2 | GenBank | Gene ID: 4929 | |
| Gene (*Homo sapiens*) | INSL4 | GenBank | Gene ID: 3641 | |
| Gene (*Homo sapiens*) | LINC00473 | GenBank | Gene ID: 90632 | |
| Gene (*Homo sapiens*) | PDE4D | GenBank | Gene ID: 5144 | |
| Cell line (*Homo-sapiens*) | A549 | ATCC | CCL-185 | |
| Cell line (*Homo-sapiens*) | H157 | ATCC | CRL-5802 | |
| Cell line (*Homo-sapiens*) | H322 | ATCC | CRL-5806 | |
| Cell line (*Homo-sapiens*) | H522 | ATCC | CRL-5810 | |
| Cell line (*Homo-sapiens*) | H2126 | ATCC | CCL-256 | |
| Cell line (*Homo-sapiens*) | H1819 | ATCC | CRL-5897 | |
| Cell line (*Homo-sapiens*) | H2087 | ATCC | CRL-5922 | |
| Cell line (*Homo-sapiens*) | H2009 | ATCC | CRL-5911 | |

*Continued on next page*

*Continued*

| Reagent type (species) or resource | Designation | Source or reference | Identifiers | Additional information |
|---|---|---|---|---|
| Cell line (*Homo-sapiens*) | H3123 | Frederic J Kaye lab | CVCL_Y295 PMID:11030152 | |
| Cell line (*Homo-sapiens*) | H23 | ATCC | CRL-5800 | |
| Cell line (*Homo-sapiens*) | H460 | ATCC | HTB-177 | |
| Cell line (*Homo-sapiens*) | H2122 | ATCC | CRL-5985 | |
| Cell line (*Homo-sapiens*) | H358 | ATCC | CRL-5807 | |
| Cell line (*Homo-sapiens*) | BEAS-2B | ATCC | CRL-9609 | |
| Cell line (*M. musculus*) | mLSCC$^{LP}$ | Francesco J DeMayo lab | PMID:31089135 | |
| Antibody | anti-CRTC1 (Rabbit Polyclonal) | Rockland Immunochemicals Inc | Cat: #600-401-936 | WB 1:1000 |
| Antibody | anti-CRTC1 (Rabbit Polyclonal) | Bethyl Laboratories | Cat: #A300-769A | ChIP 3 ug/ml |
| Antibody | anti-CRTC2 (Rabbit Polyclonal) | Bethyl Laboratories | Cat: #A300-637A | WB 1:1000 ChIP 3 ug/ml |
| Antibody | anti-CRTC3 (Rabbit Polyclonal) | Bethyl Laboratories | Cat: #A302-703A, | ChIP 3 ug/ml |
| Antibody | anti-CRTC3 (Rabbit monoclonal) | Cell Signaling Technology | Cat: #2720 | WB 1:1000 |
| Antibody | anti-LKB1 (Rabbit monoclonal) | Cell Signaling Technology | Cat: #3050 | WB 1:1000 |
| Antibody | anti-β-TUBULIN (Rabbit monoclonal) | Epitomics | Cat: #1878 | WB 1:2000 |
| Antibody | anti-HDCA1 (Rabbit Polyclonal) | Santa Cruz Biotechnology | Cat: #sc7872 | WB 1:2000 |
| Antibody | anti-β-ACTIN (Mouse monoclonal) | Sigma-Aldrich | Cat: #A5316 | WB 1:2000 |
| Recombinant DNA reagent | lentiCRISPR v2 (plasmid) | Addgene | Plasmid #52961 | |
| Recombinant DNA reagent | sgCtr- LentiCRISPRv2 (plasmid) | Addgene | Plasmid #107402 | |
| Recombinant DNA reagent | sgCRTC1- lentiCRISPR v2 (plasmid) | This paper | | sgRNA sequence cloned into lentiCRISPR v2 |
| Recombinant DNA reagent | sgCRTC2- lentiCRISPR v2 (plasmid) | This paper | | sgRNA sequence clone into lentiCRISPR v2 |
| Recombinant DNA reagent | sgCRTC3- lentiCRISPR v2 (plasmid) | This paper | | sgRNA sequence clone into lentiCRISPR v2 |
| Recombinant DNA reagent | pMSCV-GFP (plasmid) | Addgene | Plasmid #86537 | |
| Recombinant DNA reagent | pMSCV-dnCRTC (plasmid) | This paper | | dnCRTC sequence cloned into pMSCV-GFP |

*Continued on next page*

*Continued*

| Reagent type (species) or resource | Designation | Source or reference | Identifiers | Additional information |
|---|---|---|---|---|
| Recombinant DNA reagent | pcDNA FLAG TORC1 (plasmid) | Addgene | Plasmid #25718 | |
| Recombinant DNA reagent | pcDNA FLAG TORC2 (plasmid) | Addgene | Plasmid #22975 | |
| Recombinant DNA reagent | pcDNA FLAG TORC3 (plasmid) | Addgene | Plasmid #22976 | |
| Recombinant DNA reagent | lentiCas9-Blast (plasmid) | Addgene | Plasmid #52962 | |
| Recombinant DNA reagent | non-targeting control gRNA (plasmid) | Addgene | Plasmid #80180 | |
| Recombinant DNA reagent | STK11 gRNA-1 (plasmid) | Addgene | Plasmid #75912 | |
| Recombinant DNA reagent | STK11 gRNA-2 (plasmid) | Addgene | Plasmid #75913 | |
| Recombinant DNA reagent | pMD2.G (plasmid) | Addgene | Plasmid #12259 | Lentiviral Envelope |
| Recombinant DNA reagent | psPAX2 (plasmid) | Addgene | Plasmid #12260 | Lentiviral Packaging |
| Sequence-based reagent | CRTC1-qRT-F | This paper | qPCR primers | TGTCTCTCTGACC CCCTTCCAATCC |
| Sequence-based reagent | CRTC1-qRT-R | This paper | qPCR primers | GTCCGCGGGTGGT GAGAGGTA |
| Sequence-based reagent | CRTC2-qRT-F | This paper | qPCR primers | AGCCCCCTGA GTTTGCTCGC |
| Sequence-based reagent | CRTC2-qRT-R | This paper | qPCR primers | TGGGGGTAACCGC TGGTCAGT |
| Sequence-based reagent | CRTC3-qRT-F | This paper | qPCR primers | TGACCAGCAGTC CATGAGGCCA |
| Sequence-based reagent | CRTC3-qRT-R | This paper | qPCR primers | GGTCTTTGAACAG GCTGGTGCTGG |
| Sequence-based reagent | LINC00473-qRT-F | This paper | qPCR primers | AAACGCGAACG TGAGCCCCG |
| Sequence-based reagent | LINC00473-qRT-R | This paper | qPCR primers | CGCCATGCTCT GGCGCAGTT |
| Sequence-based reagent | FOS-qRT-F | This paper | qPCR primers | CACTCCAAGC GGAGACAG |
| Sequence-based reagent | FOS-qRT-R | This paper | qPCR primers | AGGTCATCAGG GATCTTGCAG |
| Sequence-based reagent | NR4A2-qRT-F | This paper | qPCR primers | GCCGGAGAGGT CGTTTGCCC |
| Sequence-based reagent | NR4A2-qRT-R | This paper | qPCR primers | AGGGTTCGCCT GGAACCTGGAA |
| Sequence-based reagent | INSL4-qRT-F | This paper | qPCR primers | GATGTGGTCCC CGATTTGGA |
| Sequence-based reagent | INSL4-qRT-R | This paper | qPCR primers | AGGTTGACACCA TTTCTTTGGG |
| Sequence-based reagent | CPS1-qRT-F | This paper | qPCR primers | CTGATGCTGCC CACACAAAC |
| Sequence-based reagent | CPS1-qRT-R | This paper | qPCR primers | AGGGGAAGGA TCGAGAAGCT |
| Sequence-based reagent | PDK4-qRT-F | This paper | qPCR primers | ACAGACAGGAA ACCCAAGCC |

*Continued on next page*

*Continued*

| Reagent type (species) or resource | Designation | Source or reference | Identifiers | Additional information |
|---|---|---|---|---|
| Sequence-based reagent | PDK4-qRT-R | This paper | qPCR primers | GTTCAACTGTT GCCCGCATT |
| Sequence-based reagent | NR4A1-qRT-F | This paper | qPCR primers | GAGTCCCAGTG GCGGAGGCT |
| Sequence-based reagent | NR4A1-qRT-R | This paper | qPCR primers | CAGGCTGCA CCCTACCCGGC |
| Sequence-based reagent | TM4SF20-qRT-F | This paper | qPCR primers | TCCAGGCTCTC TTAAAAGGTCC |
| Sequence-based reagent | TM4SF20-qRT-R | This paper | qPCR primers | ATGGTGTCGTT ACTGGTGGG |
| Sequence-based reagent | NR4A3-qRT-F | This paper | qPCR primers | GAAGAGGGCA GCCCGGCAAG |
| Sequence-based reagent | NR4A3-qRT-R | This paper | qPCR primers | ACGCAGGGCAT ATCTGGAGGGT |
| Sequence-based reagent | PTGS2-qRT-F | This paper | qPCR primers | GTTCCCACCC ATGTCAAAAC |
| Sequence-based reagent | PTGS2-qRT-R | This paper | qPCR primers | CCGGTGTTGAG CAGTTTTCT |
| Sequence-based reagent | SIK1-qRT-F | This paper | qPCR primers | AGCTTCTGAAC CATCCACACA |
| Sequence-based reagent | SIK1-qRT-R | This paper | qPCR primers | TTTGCCAGAACT TCTTCCGC |
| Sequence-based reagent | PDE4B-qRT-F | This paper | qPCR primers | CCGATCGCATTC AGGTCCTTCGC |
| Sequence-based reagent | PDE4B-qRT-R | This paper | qPCR primers | TGCGGTCTGT CCATTGCCGA |
| Sequence-based reagent | PDE4D-qRT-F | This paper | qPCR primers | AACACATGAATC TACTGGCTGA |
| Sequence-based reagent | PDE4D-qRT-R | This paper | qPCR primers | TCACACATGGG GCTTATCTCC |
| Sequence-based reagent | GAPDH-qRT-F | This paper | qPCR primers | CAATGACCCC TTCATTGACC |
| Sequence-based reagent | GAPDH-qRT-R | This paper | qPCR primers | GACAAGCTTCC CGTTCTCAG |
| Sequence-based reagent | ID1-qRT-F | This paper | qPCR primers | TTCTCCAGCA CGTCATCGAC |
| Sequence-based reagent | ID1-qRT-R | This paper | qPCR primers | CTTCAGCGAC ACAAGATGCG |
| Sequence-based reagent | LINC00473 promotor-qRT-F | This paper | qPCR primers | CTACAGACGTC ATCGCCTCC |
| Sequence-based reagent | LINC00473 promotor-qRT-R | This paper | qPCR primers | CACATTTGGGG GTGCTTGTG |
| Sequence-based reagent | NR4A2 promoter-qRT-F | This paper | qPCR primers | GGGGAAAGTG AAGTGTCG |
| Sequence-based reagent | NR4A2 promoter-qRT-R | This paper | qPCR primers | CCGCGCTCGC TTTGGTAT |
| Sequence-based reagent | sgCRTC1-A | This paper | gRNA targets | TGGCGACTTC GAACAATCCG |
| Sequence-based reagent | sgCRTC1-B | This paper | gRNA targets | TTACCCGCGCG GCCCGCGTC |
| Sequence-based reagent | sgCRTC1-C | This paper | gRNA targets | CCCAGCCGAG GCCAGTACTA |

*Continued on next page*

*Continued*

| Reagent type (species) or resource | Designation | Source or reference | Identifiers | Additional information |
|---|---|---|---|---|
| Sequence-based reagent | sgCRTC2-A | This paper | gRNA targets | GCAGCGAGAT CCTCGAAGAA |
| Sequence-based reagent | sgCRTC2-B | This paper | gRNA targets | AGGATATGTGG CGGGTGTAT |
| Sequence-based reagent | sgCRTC2-C | This paper | gRNA targets | ACAGGCCCAAAA ACTGCGAC |
| Sequence-based reagent | sgCRTC3-A | This paper | gRNA targets | CTGACGCACTGC TCCGCAGC |
| Sequence-based reagent | sgCRTC3-B | This paper | gRNA targets | AAAAAGGATATT TGTCGCCC |
| Sequence-based reagent | sgCRTC3-C | This paper | gRNA targets | AACCCGCCATCA CGGGCTGG |
| Sequence-based reagent | sg-Ctr | This paper | gRNA targets | CTTCCGCGG CCCGTTCAA |
| Commercial assay or kit | Bronchial Epithelial Cell Growth Medium kit | Lonza | Cat: # CC-4175 | BEAS-2B cell culture |
| Commercial assay or kit | Effectene Transfection Reagent | QIAGEN | Cat: #301425 | Transfection |
| Commercial assay or kit | RNeasy Mini Kit | QIAGEN | Cat: #74106 | RNA extraction |
| Commercial assay or kit | cDNA Reverse Transcription Kit | Applied Biosystems | Cat: #4368814 | |
| Commercial assay or kit | SYBR Green Supermix | Bio-Rad | Cat: #1725120 | |
| Commercial assay or kit | Alkaline Phosphatase, Calf Intestinal | New England BioLabs | Cat: #M0290 | |
| Commercial assay or kit | Nuclear and Cytoplasmic Extraction Reagents | Thermo Scientific | Cat: #78833 | |
| Commercial assay or kit | West Dura Extended Duration Substrate | Thermo Scientific | Cat: # 34076 | |
| Commercial assay or kit | GFP-Trap Magnetic Agarose | ChromoTek | Cat:# #gtma-10 | |
| Commercial assay or kit | VeriBlot for IP Detection Reagent | abcam | Cat:# ab131366 | |
| Commercial assay or kit | Dual-Luciferase Reporter Assay System | Promega | Cat:# E1910 | |
| Commercial assay or kit | FITC Annexin V Apoptosis Detection Kit | BD Bioscience | Cat: #556547 | |
| Chemical compound, drug | Hexadimethrine bromide | Sigma-Aldrich | Cat: # H9268 | polybrene |
| Chemical compound, drug | Puromycin Dihydrochloride | Gibco | Cat: #A1113803 | |
| Chemical compound, drug | Matrigel | Corning | Cat: #356231 | |

*Continued on next page*

*Continued*

| Reagent type (species) or resource | Designation | Source or reference | Identifiers | Additional information |
|---|---|---|---|---|
| Chemical compound, drug | D-Luciferin | PerkinElmer | Cat: #122799 | |
| Software, algorithm | GraphPad Prism 7 | GraphPad Prism | | |
| Software, algorithm | ImageJ software | ImageJ | | |

## Cell culture

Human NSCLC cancer cell lines (A549, H157, H322, H522, H2126, H1819, H2087, H2009, and H3123) were cultured in DMEM (Corning #10–013-CV) supplemented with 10% (vol/vol) heat-inactivated fetal bovine serum (Gibco #10437028), and penicillin (100 U/mL)/streptomycin (100 µg/mL) (HyClone #SV30010). Human NSCLC cancer cell lines (H23, H460, H2122 and H358) and mouse lung squamous carcinoma mLSCC$^{LP}$ cell line (*Liu et al., 2019*) were cultured in RPMI-1640 (HyClone # SH3002701) with 10% inactivated fetal bovine serum and penicillin/streptomycin. Immortalized human bronchial epithelial BEAS-2B cells were cultured in BEGM bronchial epithelial cell growth medium (Lonza #CC-4175). All the cells were grown at 37°C with 5% $CO_2$. The above cell lines were originally obtained from American Type Culture Collection (ATCC) or the scientists who generated the cell lines. These cell lines were not authenticated at our end, but we routinely tested the cell lines for the key gene alterations including LKB1 expression by western blotting. Mycoplasma testing is regularly performed using a MycoAlertTM Mycoplasma Detection Kit (Lonza # LT07-418), and the cell lines were free of mycoplasma in our study.

## Plasmids

The sgRNA sequences targeting CRTC1, CRTC2, and CRTC3 were designed using the CRISPR design tool (https://zlab.bio/guide-design-resources) and cloned into the lentiCRISPR v2 vector that co-expresses Cas9 (Addgene #52961) (*Sanjana et al., 2014*). The control plasmid sgCtr-Lenti-CRISPRv2 expressing a non-target sgRNA (#107402) (*Gao et al., 2017*), lentiCas9-Blast (#52962) and lentiGuide-Puro constructs containing non-targeting control gRNA (#80180), *STK11* gRNA-1 (#75912), and *STK11* gRNA-2 (#75913) were also purchased from Addgene (*Doench et al., 2016*). The sequences of gRNAs and non-targeting control were listed in *Supplementary file 1c*.

The pMSCV-dnCRTC retroviral construct was generated by cloning a DNA fragment encoding the CRTC1 CBD domain (1–55 aa) followed by a nuclear localization signal (PKKKRKV) into the backbone of the pMSCV-GFP vector (*Pui et al., 1999*) by replacing the internal ribosome entry sequence (IRES). The cAMP response element (CRE) luciferase reporter (pCRE-luc), Renilla luciferase plasmid (pEF-RL), and pFLAG-CMV2 vectors expressing individual CRTC were previously described (*Wu et al., 2005*). The pcDNA FLAG-tagged CRTC1(#22974), pcDNA FLAG-tagged CRTC2 (#22975), cDNA FLAG-tagged CRTC3 (#22976), pBABE-puro (#1764), and pBABE-FLAG-LKB1 (#8592) constructs were obtained from Addgene (*Conkright et al., 2003*; *Shaw et al., 2004*).

## CRISPR-Cas9-mediated gene knockout

LentiCRISPR constructs containing sgRNAs for CRTC1, CRTC2, or CRTC3 or control sgRNA were transfected into 293FT cells together with packaging plasmids pMD2.G and pSPAX2 using Effectene transfection reagent (Qiagen #301425). The viral supernatants were collected at 48, 72 and 96 hr after transfection. A549 cells were then infected by culture-medium-diluted viral supernatants in the presence of 6 µg/ml polybrene (Sigma #H9268) in three consecutive days and selected with puromycin (1.5 µg/ml) for 48 hr. Single-cell cloning was set up through serial dilutions in 96-well plates, followed by expansion of cell culture. The knockout clones were validated for the loss of protein expression by western blotting and for altered genomic sequences by DNA sequencing.

H322 cells were infected with lentivirus containing lentiCas9-Blast construct at 3 consecutive days as described above. The transduced cells were then selected with Blasticidin (10 µg/ml) for 48 hr.

The expression of Cas9 protein was validated by western blotting. Then the H322-Cas9 cells were infected with lentiviruses containing lentiGuide-Puro constructs expressing gRNAs for LKB1 or non-targeting control as described above. The infected cells were selected with 1.5 µg/ml puromycin for 48 hr. Elimination of LKB1 protein was then validated by western blotting.

## Retroviral transduction

293FT cells were transfected with pMSCV-dnCRTC or pMSCV-GFP constructs together with packaging plasmid pMD.MLV and pseudotyped envelope plasmid pMD2.G using Effectene transfection reagent (Qiagen #301425) as previously described (*Chen et al., 2014*). Viral supernatants were collected at 48 and 72 hr post-transfection. Targeted cells (A549, H157, H322, H522, BEAS-2B, mLSCCLP) were infected with viral supernatants mixed with fresh complete medium plus 6 µg/ml polybrene for 6 hr. Infection was performed twice in two consecutive days.

For the LKB1 addback experiment, LKB1-null A549 cells were infected with retroviruses generated by pBABE-FLAG-LKB1 or pBABE-Puro constructs twice in 2 consecutive days. The infected cells were then selected with puromycin (1.5 µg/ml) for 48 hr, and LKB1 expression was validated by western blotting.

## RT-qPCR

Total RNA was isolated using RNeasy Mini Kit (Qiagen #74106) and then reverse-transcribed into complementary DNA using a High Capacity cDNA Reverse Transcription Kit (Applied Biosystems #4368814). PCR was subsequently performed using StepOne Real-Time PCR System with iTaq Universal SYBR Green Supermix (Bio-Rad #1725120). The relative gene expression was calculated using the comparative $\Delta\Delta$Ct method. Glyceraldehyde-3-phosphate dehydrogenase (GAPDH) was used as an internal control for normalizing gene expression among different samples. The primer sequences were listed in *Supplementary file 1c*.

## Western blotting analysis

Cells were lysed in lysis buffer [10 mM Tris/Cl pH 7.5, 150 mM NaCl, 0.5 mM EDTA, 0.5% NP-40, 2 mM $Na_3VO_4$, 1 mM PMSF, 2 mg/ml protease inhibitor cocktail (cOmplete, Roche)] on ice for 30 min. Protein lysates were collected after removing insoluble fractions by centrifugation at 13,000 rpm for 15 min at 4°C. For phosphatase treatment, cells were lysed with lysis buffer without $Na_3VO_4$ and incubated with alkaline calf intestinal phosphatase (one unit per µg protein, NEB #M0290) at 37°C for 60 min. The NE-PER Nuclear and Cytoplasmic Extraction Reagents kit (Thermo Scientific #78833) was used to separate the nuclear and cytoplasmic fractions. HDAC1 and β-TUBULIN were detected as nuclear and cytoplasmic markers, respectively.

Protein lysates (~50 µg/lane) were separated on SDS-PAGE gels and electrophoretically transferred onto nitrocellulose membranes. The membranes were blocked in 5% w/v fat-free milk in TBST buffer (10 mM Tris-HCl, pH 8.0, 150 mM NaCl, 0.05% Tween 20) at room temperature for 1 hr and then incubated with primary antibodies diluted in TBST at 4°C overnight. After extensive washing with TBST, the membranes were incubated with horseradish peroxidase (HRP)-coupled secondary antibodies at room temperature for 1 hr, washed again and proteins were visualized by SuperSignal West Dura Extended Duration Substrate (Qiagen #34076).

The following antibodies were used for western blotting: anti-CRTC1 (Cat #600-401-936, Rabbit) from Rockland Immunochemicals Inc; anti-CRTC2 (Cat #A300-637A, Rabbit) and anti-PDE4D (Cat #A302-744A, Rabbit) from Bethyl Laboratories; anti-CRTC3 (Cat #2720, Rabbit), anti-LKB1 (Cat #3050, Rabbit) from Cell Signaling Technology; anti-HDCA1 (Cat # sc7872, rabbit) from Santa Cruz Biotechnology, anti-β-TUBULIN (Cat #1878, Rabbit) from Epitomics; and anti-β-ACTIN (Cat #A5316, Mouse) from Sigma-Aldrich.

## Chromatin immunoprecipitation

GFP- and dnCRTC-expressing A549 Cells were crosslinked with 1% formaldehyde for 10 min at room temperature followed by the addition of 1.25M glycine. Cells were then lysed in the lysis buffer (10 mM Tris/Cl pH 7.5; 150 mM NaCl; 0.5 mM EDTA; 0.5% NP-40; 1 mM PMSF) followed by sonication to shear DNA to 100 bp - 500 bp fragments. For the ChIP assays for the binding of dnCRTC or GFP to target gene promoters, the DNA-protein complex was then immunoprecipitated with

ChromoTek GFP-trap (ChromoTek #gtma-10) or control IgG agarose beads, and analyzed for CREB and dnCRTC by western blot. The ChIP DNA was purified for real-time PCR assays using the primers that amplify the regions spanning the CRE sites of LINC00473 and NR4A2 promoters. For ChIP assays for the binding of endogenous CRTCs to target gene promoters, the fragmented chromatins were incubated overnight with the CRTC1 antibody (Bethyl A300-769A), or CRTC2 antibody (Bethyl A300-637A), or CRTC3 antibody (Bethyl A302-703A), or control immunoglobulin G. The antibody-DNA-protein complexes were then immunoprecipitated with Protein A/G beads and the ChIP DNA was purified and analyzed as described above. The primer sequences were listed in *Supplementary file 1c*.

## Luciferase reporter assays

HEK293T cells were seeded in 24-well plates at $1 \times 10^5$ cells/well overnight and transfected with pCRE-luc firefly luciferase vector, internal control Renilla luciferase plasmid (pEF-RL), pFLAG-CMV2 vectors expressing CRTC1, or CRTC2, or CRTC3, and pMSCV-GFP or pMSCV-dnCRTC using Effectene transfection reagent (Qiagen #301425). The luciferase assays were carried out at 48 hr after transfection using a dual-luciferase assay kit (Promega, #E1910) as described previously (*Wu et al., 2005*).

## Transcriptomic analysis

Two biological replicates of RNA samples were isolated from A549 cells transduced with pMSCV-dnCRTC or pMSCV-GFP retroviruses at 72 hr post-infection and were then subjected to microarray experiment using GeneChip Human Transcriptome Array 2.0 (Affymetrix) at the Genomics Core at Sanford Burnham Research Institute. The same samples were also subjected to RNAseq analysis by Novogene. In brief, RNA-seq libraries (non-strand-specific, paired end) were prepared with the NEB-Next Ultra RNA Library Prep Kit (Illumina) and were sequenced according to the paired-end 150 bp protocol using NovaSeq 6000. The data were analyzed as previously described (*Yang et al., 2019*; *Chen et al., 2016*; *Chen et al., 2015*). Genes with an absolute fold change $\geq 2$ and an FDR p-value<0.5 were considered as significantly differentially expressed.

## Cell growth competition assay

The competitive cell growth assay was performed as previously described (*Wu et al., 2005*). In brief, lung cancer cells (A549, H157, H322, H522) were infected with pMSCV-based retroviruses expressing dnCRTC or GFP at infection rates between 40–60%. Cells were seeded and harvested every 3 days. The percentage of GFP-positive cells was determined by flow cytometry every 3 days for a total of 24 days. The percentage of GFP-positive cells at day three after the infection was considered as 100%, and the remaining data were normalized.

## Cell proliferation and apoptosis assays

Cells were seeded in the 6-well plates at $0.3 \times 10^6$ cells/well and cultured for 96 hr for cell proliferation and apoptosis assays. Cell proliferation was determined by direct counting of viable cells stained with 0.2% trypan blue solution. For apoptosis assay, cells were stained using a FITC Annexin V Apoptosis Detection Kit (BD Bioscience Cat #556547) and analyzed by Accuri C6 Flow Cytometer (BD Biosciences). Cells with Annexin V positive and PI positive or negative were considered apoptotic cells.

## Colony formation and soft agar assays

For colony formation assay, cells were grown in 6-well plates at 400 cells per well for 14 days. Each well was then fixed by fixation buffer (12.5% acetic acid/87.5% methanol) for 30 min followed by staining with 0.1% crystal violet solution (0.1% crystal violet, 10% ethanol dissolved in ddH$_2$O) for another 30 min at room temperature. Plates were then washed with tap water, air-dried, and scanned. The number of colonies from each well was counted using ImageJ. Assays were performed in triplicate.

Soft agar assays were performed in 6-well plates with 20,000 cells per well. Cells were suspended as single cell suspension in culture medium containing 0.35% noble agar (BD Biosciences, #214230) and then layered on the top of (0.5% agar in culture medium). The plates were incubated for 14

days and were stained by crystal violet solution (0.5% crystal violet-10% ethanol) at room temperature and colonies photographed under a microscope. Four images in different fields for each well were obtained. The number of colonies from each image was counted using ImageJ. Only colonies with a diameter higher than 50 µm were counted. Assays were performed in triplicate.

### Mouse xenograft assay

For subcutaneous xenograft assay, A549 and H157 cells stably expressing firefly luciferase (A549-luc/H157-luc) were infected with pMSCV-dnCRTC or pMSCV-GFP retroviruses. A total of $1 \times 10^6$ cells were diluted in 100 µl 50% Matrigel (BD Biosciences) and injected subcutaneously to dorsal flanks of 8–12 week-old NOD.SCID mice (Jackson Laboratory; stock# 001303). Tumors were measured using a vernier caliper every 1–2 days and tumor volumes were calculated using the formula: tumor volume = (length x width$^2$) x0.5. At the endpoint, mice were given 150 mg/g of D-luciferin in PBS by intraperitoneal injection for 10 min and bioluminescence was then imaged with a Xenogen In-vivo Imaging System (Caliper Life Sciences). Mice were then euthanized and tumors were dissected, photographed, weighed, fixed in 4% paraformaldehyde at 4°C for 48 hr and embedded into paraffin blocks.

For orthotopic xenograft assay, luciferase-expressing A549 and H157 cells (A549-Luc/H157 Luc) were transduced with retroviruses containing pMSCV-dnCRTC or pMSCV-GFP. A total of $2 \times 10^6$ cells were diluted in 100 µl PBS and intravenously injected into NOD/SCID mice from the tail vein. At the endpoint, bioluminescence was imaged as described above. Mice were euthanized and perfused with PBS. Lungs were then dissected out and photographed under a fluorescence stereomicroscope (Leica MZ16 F). The number of tumor nodules on the lungs in each mouse were counted under the microscope. Lungs were fixed in 4% paraformaldehyde at 4°C for 48 hr and paraffin embedded.

Paraffin tissue sections with 4 uM thickness were prepared and H and E staining and Ki-67 IHC staining were performed at the Molecular Pathology Core, University of Florida (Gainesville, FL) as previously described (*Yang et al., 2019*). Total tumor burden (tumor area/total area ×100%) was quantified from H and E sections using ImageJ.

Mouse procedures were performed following a protocol approved by the IACUC (Institutional Animal Care and Use Committee) of the University of Florida (201810386). All animals were housed, cared for, and used in an animal care facility at the University of Florida that is fully accredited by the Association for the Assessment and Accreditation of Laboratory Animal Care International (AAALAC) program in compliance with the Guide for the Care and Use of Laboratory Animals, the Animal Welfare Act and other applicable state and local regulations.

### Statistics

Data were analyzed using GraphPad Prism 7 (GraphPad Software, Inc, USA). The statistical significance was determined by two-tailed Student's *t*-test for two groups or by one-way ANOVA test for multiple groups (>2). Results were presented as the mean ± SD, and p-value<0.05 was considered statistically significant.

## Acknowledgements

We thank the ICBR Cytometry Core and Molecular Pathology Core at the University of Florida for the technical support. This work was supported by the National Institutes of Health (R01CA234351 and R01DE023641 to LW.; Z1AES103311-01 to FJD), the University of Florida Gatorade Trust (to FJK), and UF Health Cancer Center.

## Additional information

### Funding

| Funder | Grant reference number | Author |
| --- | --- | --- |
| National Cancer Institute | R01CA234351 | Lizi Wu |
| National Institute of Dental | R01DE023641 | Lizi Wu |

| | | |
|---|---|---|
| and Craniofacial Research | | |
| UF Health | | Lizi Wu |
| National Institute of Environmental Health Sciences | Z1AES103311-01 | Francesco J DeMayo |
| University of Florida | Gatorade Trust | Frederic J Kaye |

The funders had no role in study design, data collection and interpretation, or the decision to submit the work for publication.

### Author contributions

Xin Zhou, Formal analysis, Investigation, Methodology, Writing - original draft, Writing - review and editing; Jennifer W Li, Formal analysis, Methodology, Writing - review and editing; Zirong Chen, Wei Ni, Xuehui Li, Investigation, Methodology, Writing - review and editing; Rongqiang Yang, Investigation, Writing - review and editing; Huangxuan Shen, Jian Liu, Francesco J DeMayo, Jianrong Lu, Frederic J Kaye, Resources, Methodology, Writing - review and editing; Lizi Wu, Conceptualization, Formal analysis, Supervision, Funding acquisition, Methodology, Writing - original draft, Project administration, Writing - review and editing

### Author ORCIDs

Jianrong Lu (iD) http://orcid.org/0000-0002-4969-6040
Lizi Wu (iD) https://orcid.org/0000-0002-0076-2617

### Ethics

Animal experimentation: Animal studies were performed following a protocol approved by the IACUC (Institutional Animal Care & Use Committee) of the University of Florida (201810386). All animals were housed, cared for, and used in an animal care facility at the University of Florida that is fully accredited by the Association for the Assessment and Accreditation of Laboratory Animal Care International (AAALAC) program in compliance with the Guide for the Care and Use of Laboratory Animals, the Animal Welfare Act and other applicable state and local regulations.

### Decision letter and Author response

Decision letter https://doi.org/10.7554/eLife.66095.sa1
Author response https://doi.org/10.7554/eLife.66095.sa2

# Additional files

### Supplementary files

• Supplementary file 1. dnCRTC-reguated gene analysis and oligo sequences used in this study. (a) Differentially expressed genes in dnCRTC-expressing A549 lung cancer cells in comparison with GFP-expressing control cells were shown. (b) GSEA analysis revealed that multiple oncogenic signatures were negatively associated with the dnCRTC-regulated target genes. (c) Primer and sgRNA sequences used in this study.

• Transparent reporting form

### Data availability

The transcriptomic data were deposited in the NCBI GEO database GSE157722. All data generated or analyzed for this study are included in the manuscript.

The following dataset was generated:

| Author(s) | Year | Dataset title | Dataset URL | Database and Identifier |
|---|---|---|---|---|
| Wu L | 2020 | Identification of the dnCRTC-regulated genes in lung cancer cells | https://www.ncbi.nlm.nih.gov/geo/query/acc.cgi?acc=GSE157722 | NCBI Gene Expression Omnibus, GSE157722 |

The following previously published dataset was used:

| Author(s) | Year | Dataset title | Dataset URL | Database and Identifier |
|---|---|---|---|---|
| Rodón L, Svensson RU, Wiater E, Chun MH, Tsai W, Eichner LJ, Shaw RJ, Montminy M | 2019 | Genome wide screen of CREB and CRTC2 occupancy in LKB1 mutant NSCLC cell line A549 | https://www.ncbi.nlm.nih.gov/geo/query/acc.cgi?acc=GSE128871 | NCBI Gene Expression Omnibus, GSE128871 |

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
