## [Decision Letter]

**Acceptance summary:**

This work sheds important light on the role of aberrant CRTC-CREB activation in the growth of lung cancers bearing mutations in LKB1. The experiments demonstrating specific impact of pan-CRTC inhibition in suppressing LKB1-mutated lung cancer provide a rationale for the use of inhibitors of CRTC-CREB signaling as therapies for a difficult to treat molecular subtype of non-small cell lung cancer.

**Decision letter after peer review:**

Thank you for submitting your article "Dependency of LKB1-inactivated lung cancer on aberrant CRTC-CREB activation" for consideration by *eLife*. Your article has been reviewed by 3 peer reviewers, one of whom is a member of our Board of Reviewing Editors, and the evaluation has been overseen by Erica Golemis as the Senior Editor. The following individuals involved in review of your submission have agreed to reveal their identity: David Barbie (Reviewer #1); Humam Kadara (Reviewer #3).

Essential revisions:

1. In Figure 1 have the authors looked at cytoplasmic vs nuclear levels of any of the CRTC family members in LKB1 null vs wt cells? Existing models suggest that SIK mediated phosphorylation of CRTCs results in nuclear translocation. Thus, while total levels do not appear to be different between the 2 classes of cell lines, is there differential nuclear translocation of CRTC2, for example, in LKB1 null cells?

2. The most compelling data supporting specificity of their dnCRTC construct is the data in Figure 4, with unbiased RNAseq data identifying CREB target genes as the among the top downregulated genes. But this is only proven in a single cell line. Are similar target genes downregulated by qPCR in H157 cells, which they use as their other model? This is important to establish

3. Moreover, they show that these downregulated genes are upregulated in KRAS-LKB1 mutant tumors in TCGA, supporting the idea that CRTC-CREB is driving their expression uniquely in this context. But it would also be useful to look at the same subset of target genes in their KRAS-p53 mutant cell lines expressing dnCRTC (H322 and H522), to compare the relative overall expression and the degree to which dnCRTC suppresses them in the non-LKB1 mutant context.

4. Finally, isogenic LKB1 mutant cells are typically utilized to prove specificity of downstream signaling. For example if they KO LKB1 in one of their KRAS-TP53 mutant cell lines, what happens to CRTC levels/translocation and expression of these target genes? And if technically feasibly it would be most convincing if LKB1 loss specifically resensitized these cells to their dnCRTC construct.

5. In Figure 1, the authors state that expression of CRTC mRNAs is normalized to that of CRTC1. How is this being done? Is there a single probe used that recognizes all 3 CRTC isoforms? If so, this is not stated in the methods – if not, it is not possible to normalize based on PCR cycles, given the efficiency of different probes. It is also puzzling that CRTC1 appears to be much more abundant in LKB1 wt cells.

6. In Figure 1, the authors make inferences about phosphorylation – and hence activity – of CRTC isoforms in LKB-minus versus wt cells based on the migration of the protein. This is not sufficient, particularly as the Western shown indicates proteins are running in a curve. These experiments need to be supplemented by data showing migration change following phosphatase treatment of lysates, and/or by cell fractionation data showing more nuclear CRTC proteins in LKB1-minus cells.

7. The dominant negative construct dnCRTC is an essential tool for the study. The authors use ChIP to show that it associates with CREB. However, they do not show that it displaces the wt forms of CRTC from interaction with CREB. This is an important control.

8. On page 11, "we identified a list of direct dnCRTC-regulated genes, which represent an extensive set of the potential critical mediators for CRTC-CREB activation in promoting lung cancer cell growth" is a significant overstatement, and should be toned down or removed, unless validation data is provided for the importance of some of the genes identified.

9. Prior reference 42, which the authors frequently cite, note importance of overexpressing CRTC2 for activation of ID1, with validation included. What happens to ID1 expression in cells overexpressing dnCRTC?

10. On page 12, there is description of results from GSEA, but no data is shown, no statistical significance of results is provided, and the described findings are very vague. The data need to be provided and the data more accurately described, or this section should be removed.

11. Various studies have shown that LKB1-mutated lung tumors are immune muted or privileged. In fact, in the past couple of years, studies have shown that a muted immune response is a key, if not the most prominent, feature of LKB1-mutated lung cancers relative to LKB1-wild type tumors or those with other driver alterations. Indeed, LKB1-mutated tumors were shown to be non-responsive, or at best, weakly responsive to immune checkpoint blockade. A major weakness in the study is the sole dependency on immune compromised animal models. It would be important to determine the impact of dcCRTC inhibition on the immune microenvironment, in light of previous reports linking CREB, and immune infiltration/function. At least this weakness should be acknowledged and discussed in the manuscript.

12. The data in Supplementary Figure 2 showing the association of ddCRTC target genes with LKB1 mutations in NSCLC are not clear and difficult to visualize. This conclusion would benefit from statistical analysis that supports the association of dnCRTC target genes with LKB1 mutations.

13. As the authors discussed in the Introduction section, LKB1 deficiency was shown to impair salt-inducible kinases (SIK1,2, and 3) leading to phosphorylation/activation of CRTCs. It is worthwhile for the authors to discuss their finding on reduced SIK1 expression by dnCRTC (Figure 4).

14. The group discusses oncogenic gene signatures that were negatively enriched in dnCRTC-expressing versus control GFP-expressing A549 cells (page 12) but these data are not provided.

15. The H&E staining of mouse lungs colonized with A549 (Figure 7C) are not clear (strong background).

*Reviewer #1:*

In this manuscript Zhou et al. explore the role of downstream CRTC-CREB activation in KRAS-LKB1 mutant lung adenocarcinoma. First, they characterize expression of the 3 different CRTC family members across LKB1 null vs wt lung cancer cells, observing variable expression with generally higher levels of CRTC2 (previously implicated as a dependency in this setting) and CRTC3. Since KO of each family member was limited by functional reduncancy to some degree, they then engineered a dominant negative model by fusing the common CREB binding domain to GFP, and expressed this in A549 cells. Nicely, RNA-seq confirmed specific downregulation of CRTC-CREB targets in these cells, which was associated with impaired proliferation that was preferentially observed in LKB1 null cells. Finally, they demonstrate that subcutaneous implantation of dnCRTC expressing LKB1 null cells impaired tumor xenograft growth as well as metastatic lung colonization. Overall this is a well conducted study which builds upon an emerging literature highlighting the importance of SIK signaling, CRTC2, and CREB in KRAS-LKB1 mutant cancer, which still lacks effective therapy.

*Reviewer #2:*

In this article, Zhou et al. investigate the requirement for the three isoforms of CRTC (CRTC1-3) in pathogenesis of LKB1-mutant lung cancers. CRTC proteins serve as co-activators of CREB; LKB1 loss reduces activity of SIK kinases, which normally phosphorylate and cytoplasmically sequester CRTCs; in the absence of LKB1, unphosphorylated CRTCs enter the nucleus and associate with CREB. Previous work has noted that upregulation of CRTC2 is common in lung cancer, and showed elevated CRTC2 promotes lung cancer growth – however, that study found little effect of reducing CRTC2 expression. The present work investigates all three CRTCs, showing depletion of each, alone, has little effect, but simultaneous inhibition of all CRTCs through expression of a dominant negative construct that competes with CRTCs for CREB binding, but fails to activate transcription, specifically blocks the growth of LKB1-deficient but not LKB1-wt lung cancer, in vitro and in vivo. The work also suggests specific CREB transcriptional targets that may contribute to the transforming activity of CRTC-CREB. Overall, the work is carefully performed and makes a useful contribution to the field. However, some points need to be further elucidated. These include better characterization of the relative expression and modification of CRTC proteins, confirmation that the biological activity of the dominant negative construct efficiently displaces CRTCs from interaction with CREB on gene promoters, and more thorough description of the downstream transcriptional consequences of use of dominant negative CRTC.

*Reviewer #3:*

The study by Zhou and colleagues investigates the role of aberrant activation of CREB-regulated transcription co-activators (CRTCs 1, 2 and 3) and CREB in LKB1-mutated (inactivated) non-small cell lung cancer (NSCLC). Previous work has shown that LKB1-inactivated lung cancer exhibits increased CRTC-CREB activation, yet the relative contributions of the three CRTC co-activators are still not determined. Here, the study evaluated the effects of a pan-CRTC inhibitor on the malignant phenotype of LKB1-inactivated NSCLC cells and tumors. The study first shows that expression levels of the different CRTCs are variable across LKB1-wild type and-inactivated lung cancer cells and that targeting each CRTC alone does not lead to decreased cell growth and progression of LKB1-inactivated lung cancer cells. In contrast, they find that a pan-CRTC inhibitor (dnCRTC) decreased global CREB target gene expression and specifically inhibited in vitro growth of LKB1-inactivated but not LKB1-wild type lung cancer cells. The group also shows that dnCRTC significantly decreases the growth of subcutaneously xenotransplanted human LKB1-inactivated A549 and H157 lung cancer cells in NOD/SCID mice. Also, dnCRTC inhibited lung colonization of A549 and H157 cells intravenously injected in NOD-SCID mice. The study is in general carefully designed and clearly shows the significance of aberrant CRTC-CREB signaling in the growth of LKB1-inactivated lung cancer cells.

---

## [Author Response]

Essential revisions:1. In Figure 1 have the authors looked at cytoplasmic vs nuclear levels of any of the CRTC family members in LKB1 null vs wt cells? Existing models suggest that SIK mediated phosphorylation of CRTCs results in nuclear translocation. Thus, while total levels do not appear to be different between the 2 classes of cell lines, is there differential nuclear translocation of CRTC2, for example, in LKB1 null cells?

Thank you for the comment. We have performed new experiments to validate that LKB1 loss results in CRTC de-phosphorylation and nuclear translocation, two essential steps for CRTC activation of CREB-mediated transcription. Specifically, we assessed the phosphorylation status and subcellular localization of the three CRTC proteins by performing phosphatase treatment and subcellular fractionation followed by immunoblot analysis. As shown in Figure 1—figure supplement 1A, endogenous CRTC1, CRTC2, and CRTC3 proteins showed slow migration patterns in LKB1-expressing lung cancer cells (H322). Upon treatment of phosphatase, the mobility of the endogenous CRTC proteins in LKB1-expressing lung cancer cells was shifted to the underphosphorylated forms, which matched the mobility patterns of CRTCs in the LKB1-null lung cancer cells. These data demonstrate that endogenous CRTCs are predominantly phosphorylated in LKB1-expressing cells and dephosphorylated in LKB1-null cells. Further immunoblot analysis of the nuclear and cytoplasmic fractions revealed that CRTC proteins were predominantly detected as dephosphorylated, nuclear forms in LKB1-null cells and phosphorylated, cytoplasmic forms in LKB1-expressing cells (Figure 1—figure supplement 1B). Reintroduction of LKB1 to LKB1null A549 cells led to an increase in the levels of phosphorylated CRTCs, which correlated with a decrease in nuclear CRTCs and an increase in cytoplasmic CRTCs (Figure 1—figure supplement 2). Further, LKB1 knockout in LKB1-expressing H322 cells caused an increase in dephosphorylated, nuclear forms of CRTCs and a decrease in phosphorylated, cytoplasmic forms of CRTCs (Figure 1—figure supplement 3). All these data validate LKB1 regulation of CRTC phosphorylation and subcellular localization, further supporting the existing model where LKB1-dependent SIKs mediate phosphorylation and cytoplasmic retention of CRTCs. Therefore, there is differential nuclear translocation of CRTCs in LKB1deficient cells in comparison to LKB1-wt cells.

2. The most compelling data supporting specificity of their dnCRTC construct is the data in Figure 4, with unbiased RNAseq data identifying CREB target genes as the among the top downregulated genes. But this is only proven in a single cell line. Are similar target genes downregulated by qPCR in H157 cells, which they use as their other model? This is important to establish

We performed RT-qPCR in a second LKB1-null cancer cell line, H157, to assess if dnCRTC also affects the expression levels of the 12 dnCRTC-regulated genes that we identified from the gene profiling analysis of dnCRTC-expressing vs control A549 cell. We also analyzed ID1, a gene recently identified as the CRTC2/CREB-regulated target. As shown in Figure 4—figure supplement 1, all the analyzed genes except for *INSL4* and *PDK4* were also downregulated in H157 cells. It should be noted that there are different basal expression levels of the genes that were tested, and *INSL4* and *PDK4* both had a very low basal expression in the H157 cell line. These data indicate that dnCRTC is capable of inhibiting a similar set of target genes in LKB1-null lung cancer.

3. Moreover, they show that these downregulated genes are upregulated in KRAS-LKB1 mutant tumors in TCGA, supporting the idea that CRTC-CREB is driving their expression uniquely in this context. But it would also be useful to look at the same subset of target genes in their KRAS-p53 mutant cell lines expressing dnCRTC (H322 and H522), to compare the relative overall expression and the degree to which dnCRTC suppresses them in the non-LKB1 mutant context.

Thank you for the suggestions. We performed RT-qPCR to determine the effects of dnCRTC expression on the same subset of target genes in the LKB1-wt cancer cell lines, H322 and H522. As shown in the new Figure 4—figure supplement 1, expression of dnCRTC had no significant effects on the expression of the target genes tested. It is noted that the majority of these dnCRTC-regulated target genes show marked up-regulation in LKB1-null cancer cells (A549 and H157) compared with LKB1-wt cells (H322 and H522).

4. Finally, isogenic LKB1 mutant cells are typically utilized to prove specificity of downstream signaling. For example if they KO LKB1 in one of their KRAS-TP53 mutant cell lines, what happens to CRTC levels/translocation and expression of these target genes? And if technically feasibly it would be most convincing if LKB1 loss specifically resensitized these cells to their dnCRTC construct.

Thank you for the suggestions. We established LKB1-positive H322 cells that were transduced with lentiviruses containing non-targeting gRNA (gNT), or gRNA targeting LKB1 and performed phosphatase treatment and subcellular fractionation experiments. The results showed that LKB1 knockout resulted in enhanced dephosphorylated CRTCs and increased nuclear CRTC levels (Figure 1—figure supplement 3). We also observed that LKB1 knockout led to a significant up-regulation of multiple target genes by RT-qPCR assays, although not to the extent that was observed in the naturally occurring human LKB1-null lung cancer cells (Figure 4—figure supplement 2). Consistent with effects on the target genes, we did not observe changes in cell proliferation in H322-LKB1 KO cells (not shown).

Since malignant phenotypes of these lung cancer H322 cells are not driven by the loss of LKB1 and not dependent on dnCRTC (Figure 5), it remains unclear whether they can become sensitive to dnCRTC. One possibility for the mild observed effects after LKB1 knockout is that the target genes could be regulated by other factors depending on cellular context, which would require further studies.

5. In Figure 1, the authors state that expression of CRTC mRNAs is normalized to that of CRTC1. How is this being done? Is there a single probe used that recognizes all 3 CRTC isoforms? If so, this is not stated in the methods – if not, it is not possible to normalize based on PCR cycles, given the efficiency of different probes. It is also puzzling that CRTC1 appears to be much more abundant in LKB1 wt cells.

Sorry that we did not explain our analysis clearly. We used individual primer sets for detecting *CRTC1*, *CRTC2*, and *CRTC3* and normalized their expression against an internal control, the housekeeping gene GAPDH. For the analysis, we first normalized all three CRTC transcript levels against the level of the GAPDH transcript, individually. For the data presentation, we assigned the expression level of CRTC1 in BEAS-2B as 1, and presented the expression levels for the three CRTCs in various cell lines as relative values to that of CRTC1 in BEAS-2B. We have provided the original data and further explained the analysis in the Figure legend. We agree with the reviewer that different primer sets could have different PCR efficiency, and we did our best to consider this possibility when we designed the primers. The data from this experiment only provides us a general idea of the relative expression levels of three CRTC genes.

6. In Figure 1, the authors make inferences about phosphorylation – and hence activity – of CRTC isoforms in LKB-minus versus wt cells based on the migration of the protein. This is not sufficient, particularly as the Western shown indicates proteins are running in a curve. These experiments need to be supplemented by data showing migration change following phosphatase treatment of lysates, and/or by cell fractionation data showing more nuclear CRTC proteins in LKB1-minus cells.

Thank you for the suggestions. This is a similar comment as Comment #1. In brief, we performed CIP treatment and subcellular fractionations followed by immunoblotting analysis of three CRTCs using LKB1-null and LKB1-expressing cells, LKB1-addback cells, and LKB1 knockout cells. Our new results (Figure 1—figure supplement 1-3) demonstrate that enhanced levels of nuclear, dephosphorylated CRTCs in LKB1-null cancer cells.

7. The dominant negative construct dnCRTC is an essential tool for the study. The authors use ChIP to show that it associates with CREB. However, they do not show that it displaces the wt forms of CRTC from interaction with CREB. This is an important control.

Thank you for the comment. Since the interaction of CRTC and CREB requires DNA (Luo et al., PNAS 2012, PMID: 23213254), we evaluated the effect of dnCRTC expression on the enrichment of endogenous CRTC proteins in the dnCRTC-target gene promoters by ChIP assays. Specifically, we crosslinked A549-GFP vs A549-dnCRTC cells and extracted chromatin for immunoprecipitation with anti-CRTC1 (Bethyl A300-769A), anti-CRTC2 (Bethyl # A300-637A), anti-CRTC3 antibodies (Bethyl A302-703A) or control IgG. The ChIP DNA was analyzed by RT-qPCR to detect the target gene promoter sequences that encompass the cAMP responsive element (CRE) site. As shown in the new Figure 3G, there was a significant reduction in the levels of endogenous CRTC1-3 proteins that were associated with the promoters of LINC00473 and NR4A2 genes that encompass the CRE sites. These data demonstrate that dnCRTC displaces endogenous CRTC1-3 proteins from the target gene promoters in LKB1-null lung cancer cells.

8. On page 11, "we identified a list of direct dnCRTC-regulated genes, which represent an extensive set of the potential critical mediators for CRTC-CREB activation in promoting lung cancer cell growth" is a significant overstatement, and should be toned down or removed, unless validation data is provided for the importance of some of the genes identified.

Several dnCRTC-regulated target genes identified in this study have been previously studied by our group and others, which included LINC00473 (Chen et al. J Clin Invest 2016, PMID 2714039), INSL4 (Yang et al. J Natl Cancer Inst. 2019, PMID 30423141), CPS1 (Kim et al. Nature 2017 PMID 28538732; Celiktas et al. J Natl Cancer Inst. 2017 PMID: 28376202), NR4A2 (Oncogene 2010 PMID:20010869), and PTGS2 (Cao et al. J Natl Cancer Inst. 2015 PMID: 25465874). The expression of these genes was found to be up-regulated in LKB1inactivated lung cancer cell lines and patient tumors. The down-regulation of these genes individually reduced the growth and survival of LKB1-inactivated lung cancers, indicating that they are important in mediating the loss of LKB1 tumor suppression.

9. Prior reference 42, which the authors frequently cite, note importance of overexpressing CRTC2 for activation of ID1, with validation included. What happens to ID1 expression in cells overexpressing dnCRTC?

In our gene expression profiling experiment, *ID1* was down-regulated but did not meet the cutoff criteria we used (FC>=2.0, p <0.05), as *ID1* expression had a fold change of -1.72 (FDR p<0.05) in dnCRTC-expressing A549 cells as compared with the control cells.

We further analyzed that ID1 expression by RT-qPCR and observed that *ID1* expression was also downregulated by dnCRTC in two LKB1-null lung cancer cell lines (A549 and H157), but not in two LKB1-expressing lung cancer cells (H322 and H522). Interestingly, we noted that the levels of *ID1* transcripts were similar in 4 cell lines we tested. The data are now presented in Figure 4—figure supplement 1.

10. On page 12, there is description of results from GSEA, but no data is shown, no statistical significance of results is provided, and the described findings are very vague. The data need to be provided and the data more accurately described, or this section should be removed.

We have included the data from GSEA analysis in Supplementary file 1b (the former Supplemental Table 2 is now shown in Supplementary file 1c).

11. Various studies have shown that LKB1-mutated lung tumors are immune muted or privileged. In fact, in the past couple of years, studies have shown that a muted immune response is a key, if not the most prominent, feature of LKB1-mutated lung cancers relative to LKB1-wild type tumors or those with other driver alterations. Indeed, LKB1-mutated tumors were shown to be non-responsive, or at best, weakly responsive to immune checkpoint blockade. A major weakness in the study is the sole dependency on immune compromised animal models. It would be important to determine the impact of dcCRTC inhibition on the immune microenvironment, in light of previous reports linking CREB, and immune infiltration/function. At least this weakness should be acknowledged and discussed in the manuscript.

Thank you for raising this very important point. We agree with the reviewer about the importance of immune characteristics associated with LKB1-inactivated lung cancer since LKB1 mutations were shown to confer immunosuppressive tumor microenvironment and the resistance to immune checkpoint blockade in preclinical models and clinical trials. This current study focused on addressing the impact of dnCRTC expression on cancer cells using human lung cancer cell lines and xenograft tumors in immunocompromised mice, which did not allow us to investigate its impact on tumor immune microenvironment. We have now acknowledged this limitation in the discussion (Page 19, Line 433-439). For the future study, we plan to generate mouse models of lung cancer driven by oncogenic KRAS and loss of LKB1, in combination with expression of dnCRTC, to determine the effects of dnCRTC on tumor progression, tumor microenvironment, and therapy.

12. The data in Supplementary Figure 2 showing the association of ddCRTC target genes with LKB1 mutations in NSCLC are not clear and difficult to visualize. This conclusion would benefit from statistical analysis that supports the association of dnCRTC target genes with LKB1 mutations.

Thank you for the suggestion. We have downloaded the data from TCGA human lung adenocarcinoma (LUAD) and lung squamous cell carcinoma (LSCC) from cbioportal.org. and performed statistical analysis to compare expression levels of the 12 top dnCRTCregulated target genes, including INSL4, CPS1, NR4A2, PDK4, LINC00473, NR4A1, TM4SF20, NR4A3, PTGS2, SIK1, PDB4B and PDE4D in LKB1-wt and LKB1-mut cohorts.

Our results revealed high expression of the majority of these target genes in LKB1-mutant vs LKB1-wt tumors. We have presented these results as Figure 4—figure supplement 3 and the original Supplementary Figure 2 was removed.

13. As the authors discussed in the Introduction section, LKB1 deficiency was shown to impair salt-inducible kinases (SIK1,2, and 3) leading to phosphorylation/activation of CRTCs. It is worthwhile for the authors to discuss their finding on reduced SIK1 expression by dnCRTC (Figure 4).

Thank you for the suggestion. We have added several sentences to discuss the reduced SIK1 by dnCRTC (Page 18, Line 413-416). “It should be noted that SIK1 is a transcriptional target induced by cAMP-CREB signaling (53) and it was up-regulated in LKB1-null cancer cells (34). SIK1 down-regulation by dnCRTC was only observed in LKB1-null, but not LKB-wt cancer cells in this study (Figure 4—figure supplement 1), which is unlikely to have functional consequences as SIK1 kinase is impaired in LKB1-null cancer cells.”

14. The group discusses oncogenic gene signatures that were negatively enriched in dnCRTC-expressing versus control GFP-expressing A549 cells (page 12) but these data are not provided.

Thank you for pointing this out and we have now included the data in new Supplementary file 1b.

15. The H&E staining of mouse lungs colonized with A549 (Figure 7C) are not clear (strong background).

File conversion might affect our image qualities and we have now provided images with higher resolution.